# Optimal and Approximate Adaptive Stochastic Quantization

**Ran Ben Basat**
UCL

**Yaniv Ben-Itzhak**
VMware Research

**Michael Mitzenmacher**
Harvard University

**Shay Vargaftik**
VMware Research

## Abstract

Quantization is a fundamental optimization for many machine learning (ML) use cases, including compressing gradients, model weights and activations, and datasets. The most accurate form of quantization is adaptive, where the error is minimized with respect to a given input rather than optimizing for the worst case. However, optimal adaptive quantization methods are considered infeasible in terms of both their runtime and memory requirements.

We revisit the Adaptive Stochastic Quantization (ASQ) problem and present algorithms that find optimal solutions with asymptotically improved time and space complexities. Our experiments indicate that our algorithms may open the door to using ASQ more extensively in a variety of ML applications. We also present an even faster approximation algorithm for quantizing large inputs on the fly.

## 1 Introduction

Quantization is central to optimizing a large range of machine learning (ML) applications. It is often used for compressing gradients to reduce network requirements in distributed and federated learning (e.g., [1, 2, 3, 4, 5, 6]); for quantization of datasets for faster training and inference (e.g., [7]); and for reducing the memory footprint while accelerating the computation for large models' inference via post-training quantization (e.g., [8, 9]) and quantization-aware training (e.g., [10, 11]) of model weights, activations and key-value (KV) caches [12].

A fundamental quantization method is *stochastic quantization*, where one quantizes an input vector $X \in \mathbb{R}^d$ to $\widehat{X} \in Q^d$ using a set $Q \subset \mathbb{R}$ of $|Q| = s$ quantization values so that each entry is unbiased [13]. That is, each $x \in X$ is (randomly) quantized to a value $\widehat{x} \in Q$ such that $\mathbb{E}[\widehat{x}] = x$.

Previous unbiased quantization works considered different approaches. Some are distribution-agnostic, i.e., design the quantization without optimizing it for the specific input. For example, [1, 14, 15] set quantization values with respect to global properties such as the vector's norm, or minimum and maximum values.

Other works, e.g., [1, 3, 4, 16, 17, 18, 19], optimize for the worst case $X$ by applying a reversible transformation (e.g., the randomized Hadamard transform) before quantization that converts it into a vector $X'$ with a controlled distribution (e.g., with $\max(X') - \min(X') = \tilde{O}(\|X\|_2 / \sqrt{d})$). The decoder then applies the inverse transformation on the quantized $X'$ to obtain an estimate of $X$.

In contrast, some solutions use the fact that, in many cases, the inputs to be quantized have a significant structure that can be leveraged to reduce the quantization error. For example, DNN gradients (which are often compressed in distributed and federated learning applications to reduce bandwidth [20, 21]) were observed to follow LogNormal-like [22] or Normal-like [23, 24] distributions. As another example, the distribution of deep activation layers appears to follow a sub-Weibull distribution [25].

38th Conference on Neural Information Processing Systems (NeurIPS 2024).

To alleviate the need to assume an input distribution, the Adaptive Stochastic Quantization (ASQ) problem (e.g., [26, 27, 28]) considers selecting $Q$ adaptively, i.e., with respect to the specific input $X$, that minimizes the mean squared error (MSE, also known as the sum of variances) given by

$$\mathbb{E}\left[\left\|\widehat{X} - X\right\|_2^2\right] = \sum_{x \in X} \text{Var}[\widehat{x}] \ ,$$

where $\widehat{X} = \{\widehat{x} \mid x \in X\}$ is the vector of quantized values.

Unfortunately, known ASQ solutions are not practical for the large-size vectors that commonly appear in ML applications. One aspect of the problem's difficulty is that it is known to be non-convex even for $s = 4$ (two-bit quantization) [28], which excludes many natural solution methods such as gradient descent. ZipML [26] approaches the challenge using a dynamic programming approach that allows one to optimize $Q$ in polynomial time. However, this solution has a significant overhead and solving the problem optimally is often considered to be impractical; for example, [28] states

*"To find the optimal sequence of quantization values, a dynamic program is solved whose computational and memory cost is quadratic ... For this reason, ZipML is impractical for quantizing on the fly"*.

As another evidence of the problem's hardness, previous work [27] solves the problem only for a given (Weibull) distribution, writing that

*"The empirical distribution is usually non-differentiable, making the searching of Q infeasible"*.

Nevertheless, there is significant interest in advancing ASQ solutions towards wider adoption as even approximate adaptive solutions like ALQ [28] have been shown to have lower MSE than advanced distribution-agnostic methods such Non-Uniform QSGD (NUQSGD) [29]. ASQ methods can also improve more complex schemes (e.g., including the aforementioned that utilize worst-case to average-case transformations) by replacing distribution-agnostic quantization with an adaptive one.

In this paper, we show that one can, in fact, solve the ASQ problem optimally and efficiently. To this end, we introduce QUIVER, an algorithm that features novel acceleration methods and leverages the structure of the underlying problem to reduce the runtime complexity from $O(s \cdot d^2)$ to $O(s \cdot d)$ and the space complexity from $O(d^2)$ to $O(s \cdot d)$.

This improvement arises from the observation that the optimal solution, for given input parameters $s, d$, can be efficiently derived from the solutions for $\{s - 1, d' \mid d' \in \{2, 3, \ldots, d\}\}$ by a reduction to the problem of finding the row maximas in an *implicitly* defined totally monotone matrix. This problem is known to have fast algorithms assuming that, for any $1 \leq k \leq j \leq d$, the sum of variances of points $\{x_k, \ldots, x_j\}$ can be computed in constant time when quantized to $\{x_k, x_j\}$, a property that is achieved by our new preprocessing method.

We then further accelerate QUIVER by deriving a closed-form solution for $s = 3$. In turn, this yields a faster solution for any $s$, by a variant of QUIVER that places two quantization values at a time instead of one. Finally, by discretizing the search space for $Q$, we show a fast approximation variant of QUIVER. This variant introduces an appealing tradeoff between accuracy and speed, making it suitable for quantizing large vectors on the fly.

We implement our algorithms in C++ and demonstrate their efficiency. For example, on a commodity PC, QUIVER can compute the *optimal* 4-bit quantization values ($s = 16$) for a vector with $d = 1M$ entries in under a second and compute an accurate approximation in just six milliseconds. We evaluate our solutions compared to state-of-the-art ASQ methods on a variety of distributions considering different vector sizes and number of quantization values and demonstrate a speedup of up to four orders of magnitude. We open source the code of the paper [30].

We note that there are many works that investigate different forms of compression, including non-adaptive quantization (e.g., QSGD [14]), biased quantization (e.g., top-$k$ [31]), sparsification (e.g., [32]), sparse coding (e.g., [33]), low-rank decomposition (e.g., PowerSGD [34]), variable-length coding (e.g., EDEN [4]) and more. Many of these are orthogonal to our work and can be used in conjunction with it. For example, one can use ASQ to quantize a sparsified or transformed vector or apply variable-length encoding to further reduce the size of the quantized vector.

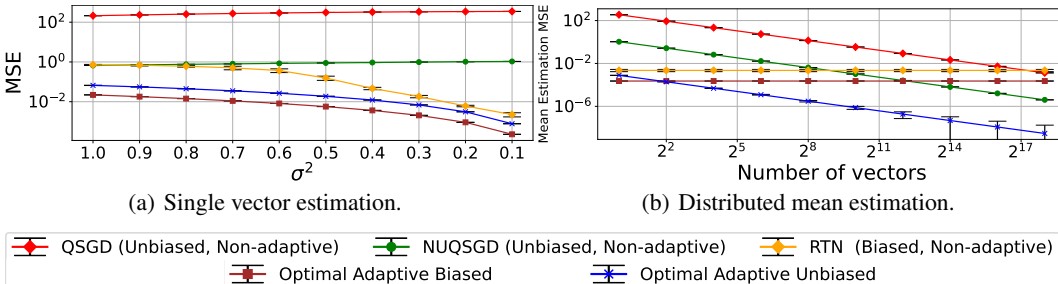

(a) Single vector estimation.  (b) Distributed mean estimation.

- ◆ QSGD (Unbiased, Non-adaptive)    ● NUQSGD (Unbiased, Non-adaptive)    ◆ RTN (Biased, Non-adaptive)
- ■ Optimal Adaptive Biased    ✳ Optimal Adaptive Unbiased

Figure 1: An experiment with dimension $d = 10M$ and $s = 10$ quantization values. Figure 1(a) shows the empirical MSE of quantizing a single vector with i.i.d. $\mathrm{LogNormal}(0, \sigma^2)$ entries. It shows that adaptive methods are more accurate than non-adaptive and that the optimal biased method is more accurate than the optimal unbiased one. However, as shown in Figure 1(b), for distributed mean estimation, the bias may not cancel out when averaging quantized inputs (here, we used a standard setup where all vectors are identical, e.g., see [17], with i.i.d. $\mathrm{LogNormal}(0, 1/2)$ distributed entries) and the advantage of unbiased methods accordingly increases with the number of inputs. Each data point is averaged over ten runs with the standard deviation reported.

## 2 Background

### 2.1 Motivation

We now briefly explain the benefits of ASQ compared to alternative methods.

**The benefits of adaptivity** Unbiased solutions such as QSGD [14] and NUQSGD [29] rely only on global properties (e.g., the input's norm) when selecting $Q$. Figure 1(a) shows the benefit of adaptivity by illustrating the potential MSE reduction from selecting $Q$ optimally for the specific input. A similar behavior is observed for biased methods where the non-adaptive Round-To-Nearest (RTN) has a higher error than the optimal adaptive biased scalar quantizer, $k$-means. As shown, this can translate to orders of magnitude lower error, depending on the data's skew.

**The benefits of unbiasedness** In many cases, it is beneficial for the quantization to be unbiased. For example, when there are $n$ senders (e.g., when doing distributed mean estimation [1, 2, 4, 17, 18]), having unbiased and independent estimates of the vectors allows the mean estimation's MSE to decay proportionally to $\frac{1}{n}$; with biased quantization, the MSE may not decay with respect to $n$ since the errors may be correlated [17] (e.g., when all clients have the same vector). This benefit is demonstrated in Figure 1(b), which shows that while biased adaptive solutions have lower error for a small number of vectors (1-2), having unbiased quantization is critical to lowering the error for a large $n$.

As another example, it was recently shown that compressing large language model parameters with biased techniques such as RTN may result in inferior performance than uniform stochastic quantization [35]. This outcome arises because the LLM layers' parameters are used to compute inner products with their inputs. Having these inner products themselves be unbiased leads to smaller errors in layers' outputs, which in turn leads to better performance.

### 2.2 Preliminaries

Given two quantization values $a, b$ and a number $x \in [a, b]$, Stochastic Quantization (SQ) is a procedure that rounds $x$ to $\widehat{x}$ where $\widehat{x} \in \{a, b\}$. Specifically, $\widehat{x}$ obtains the value $a$ with probability $p_a = \frac{b-x}{b-a}$ and the value $b$ otherwise, i.e., with probability $p_b = 1 - p_a = \frac{x-a}{b-a}$. An important property of SQ is that the expected rounded value is *unbiased*, i.e., $\mathbb{E}[\widehat{x}] = a \cdot p_a + b \cdot p_b = x$. The variance of stochastically quantizing $x$ is then given by $\mathbb{E}[(x - \widehat{x})^2] = (x-a)^2 \cdot p_a + (x-b)^2 \cdot p_b = (b-x)(x-a)$. Given a vector $X \in \mathbb{R}^d$ and an integer $s \geq 2$, the Adaptive Stochastic Quantization (ASQ) problem [26, 27, 28] looks for a set of quantization values $Q$ where $|Q| \leq s$ and $Q$ minimizes the mean squared error (MSE) that results from rounding $X$ to $\widehat{X} \in Q^d$ by stochastically quantizing each entry $x \in X$ with values $a_x = \max\{q \in Q \mid q \leq x\}$ and $b_x = \min\{q \in Q \mid q \geq x\}$.

Formally, ASQ seeks to minimize the MSE, given by $\mathbb{E}[\|X - \widehat{X}\|_2^2] = \sum_{x \in X}(b_x - x)(x - a_x)$, where $\mathbb{E}[\widehat{X}] = X$ holds by construction.

## 2.3 Existing ASQ methods

Leveraging the fact that there exists an optimal solution in which $Q \subseteq X$ [26] (i.e., the quantization values are a subset of the input), one can naively solve the problem in $d^{\Theta(s)}$ time by going over all choices for the quantization values. Instead, the following dynamic program (DP) allows us to solve it optimally and in polynomial time for any $s$ [26]. Given a *sorted* vector $X = \langle x_1, \ldots, x_d \rangle$, we denote by $MSE[i, j]$ the optimal MSE of quantizing the prefix vector $X_j = \langle x_1, \ldots, x_j \rangle$ using $i$ quantization values *that include* $x_j$, that is:

$$MSE[i, j] = \min_{Q:|Q| \leq i, x_j \in Q} \sum_{x \in X_j} (b_x - x)(x - a_x).$$

Our goal is to compute a set of quantization values $Q$ that results in an optimal MSE of $MSE[s, d]$. Accordingly, we express the dynamic program as follows. We first define $C[k, j]$ as the sum of variances of all vector entries in the range $[x_k, x_j]$ where $x_k, x_j \in Q$ are two consecutive quantization values, i.e., $C[k, j] = \sum_{x \in [x_k, x_j]} (x_j - x)(x - x_k)$. Here and when clear from context, to simplify notation, we write $\sum_x$ to denote $\sum_{x \in X}$.

For $i \in \{2, \ldots, s\}$, $j \in \{i, \ldots, d\}$, we set $MSE[2, j] = C[1, j] \ \forall j$ and use the recurrence

$$MSE[i, j] = \min_{k \in \{i, \ldots, j\}} MSE[i - 1, k] + C[k, j].$$

Here, the index $k$ denotes the entry in $X$, $x_k$, of the rightmost quantization value to the left of $x_j$. A naive solution for the above DP is first to compute the matrix $C$ (which takes $O(d^3)$ time and $O(d^2)$ space) and then calculate $MSE[i, j]$ for all $i, j$, and thus $Q$, in $O(s \cdot d^2)$ time and $O(s \cdot d)$ space. In Appendix A, we describe a simple algorithm that implements this dynamic program.

An improved solution, ZipML [26], uses $O(s \cdot d^2)$ time and $O(d^2)$ space, but it remains infeasible even for moderate (e.g., $d = 10^5$) dimensions. Accordingly, we next design novel techniques to asymptotically improve both the space and time complexities.

## 3 Optimization Using Pre-processing

The first ingredient in our solution is the usage of preprocessed arrays that allow us to efficiently compute $C[k, j]$ in constant time, at the cost of only $O(d)$ additional space. We define the following arrays, $\beta, \gamma \in \mathbb{R}^d$, that store the cumulative sums of the vector and its squared entries:

$$\beta_j = \sum_{x \in X_j} x \quad , \quad \gamma_j = \sum_{x \in X_j} x^2 \qquad \forall j \in \{1, \ldots, d\} \ .$$

Denoting $\beta_0 = \gamma_0 = 0$, both are computable in $O(d)$ time as $\beta_j = \beta_{j-1} + x_j$ and $\gamma_j = \gamma_{j-1} + x_j^2$.

We can then express $C[k, j]$ as follows:

$$\begin{aligned}
C[k, j] &= \sum_{x \in [x_k, x_j]} (x_j - x)(x - x_k) = \sum_{x \in (x_k, x_j]} (x_j - x)(x - x_k) \\
&= -x_j \cdot x_k \cdot \sum_{x \in (x_k, x_j]} 1 + (x_j + x_k) \cdot \sum_{x \in (x_k, x_j]} x - \sum_{x \in (x_k, x_j]} x^2 \\
&= -x_j \cdot x_k \cdot (j - k) + (x_j + x_k) \cdot (\beta_j - \beta_k) - (\gamma_j - \gamma_k).
\end{aligned}$$

With this optimization, we can evaluate $C[k, j]$ in constant time, yielding a solution that uses $O(s \cdot d)$ memory instead of $O(d^2)$. Next, we show how to improve the runtime.

## 4 The QUIVER Algorithm

To derive a faster algorithm, we observe that $C$ satisfies the quadrangle inequality, defined below:

**Definition 4.1.** A function $w\colon \{1, \ldots, d\} \times \{1, \ldots, d\} \to \mathbb{R}$ satisfies the quadrangle inequality if for any $\mathsf{a} \leq \mathsf{b} \leq \mathsf{c} \leq \mathsf{d}$: $\ w[\mathsf{a}, \mathsf{c}] + w[\mathsf{b}, \mathsf{d}] \leq w[\mathsf{a}, \mathsf{d}] + w[\mathsf{b}, \mathsf{c}]$.

**Lemma 4.2.** $C$ *satisfies the quadrangle inequality.*

*Proof.* We first observe that for any $x \in [x_{\mathsf{a}}, x_{\mathsf{b}}]$ :

$$(x_{\mathsf{c}} - x)(x - x_{\mathsf{a}}) = (x_{\mathsf{d}} - x)(x - x_{\mathsf{a}}) + (x_{\mathsf{c}} - x_{\mathsf{d}})(x - x_{\mathsf{a}}) \leq (x_{\mathsf{d}} - x)(x - x_{\mathsf{a}}). \tag{1}$$

For any $x \in [x_{\mathsf{c}}, x_{\mathsf{d}}]$, we similarly get:

$$(x_{\mathsf{d}} - x)(x - x_{\mathsf{b}}) = (x_{\mathsf{d}} - x)(x - x_{\mathsf{a}}) + (x_{\mathsf{d}} - x)(x_{\mathsf{a}} - x_{\mathsf{b}}) \leq (x_{\mathsf{d}} - x)(x - x_{\mathsf{a}}). \tag{2}$$

Similarly, for $x \in [x_{\mathsf{b}}, x_{\mathsf{c}}]$, we have that:

$$\begin{aligned}
(x_{\mathsf{c}} - x)(x - x_{\mathsf{a}}) + (x_{\mathsf{d}} - x)(x - x_{\mathsf{b}}) &= (x_{\mathsf{c}} - x)(x - x_{\mathsf{b}}) + (x_{\mathsf{d}} - x)(x - x_{\mathsf{a}}) + (x_{\mathsf{a}} - x_{\mathsf{b}})(x_{\mathsf{d}} - x_{\mathsf{c}}) \\
&\leq (x_{\mathsf{c}} - x)(x - x_{\mathsf{b}}) + (x_{\mathsf{d}} - x)(x - x_{\mathsf{a}}). 
\end{aligned} \tag{3}$$

Therefore, we get:

$$\begin{aligned}
C[\mathsf{a}, \mathsf{c}] + C[\mathsf{b}, \mathsf{d}] \quad &= \sum_{x \in [x_{\mathsf{a}}, x_{\mathsf{c}}]} (x_{\mathsf{c}} - x)(x - x_{\mathsf{a}}) + \sum_{x \in [x_{\mathsf{b}}, x_{\mathsf{d}}]} (x_{\mathsf{d}} - x)(x - x_{\mathsf{b}}) \\
&= \sum_{x \in [x_{\mathsf{a}}, x_{\mathsf{b}}]} (x_{\mathsf{c}} - x)(x - x_{\mathsf{a}}) + \sum_{x \in [x_{\mathsf{c}}, x_{\mathsf{d}}]} (x_{\mathsf{d}} - x)(x - x_{\mathsf{b}}) + \sum_{x \in [x_{\mathsf{b}}, x_{\mathsf{c}}]} (x_{\mathsf{c}} - x)(x - x_{\mathsf{a}}) + (x_{\mathsf{d}} - x)(x - x_{\mathsf{b}}) \\
&\leq \sum_{x \in [x_{\mathsf{a}}, x_{\mathsf{b}}]} (x_{\mathsf{d}} - x)(x - x_{\mathsf{a}}) + \sum_{x \in [x_{\mathsf{c}}, x_{\mathsf{d}}]} (x_{\mathsf{d}} - x)(x - x_{\mathsf{a}}) + \sum_{x \in [x_{\mathsf{b}}, x_{\mathsf{c}}]} (x_{\mathsf{c}} - x)(x - x_{\mathsf{b}}) + (x_{\mathsf{d}} - x)(x - x_{\mathsf{a}}). \\
&= \sum_{x \in [x_{\mathsf{a}}, x_{\mathsf{d}}]} (x_{\mathsf{d}} - x)(x - x_{\mathsf{a}}) + \sum_{x \in [x_{\mathsf{b}}, x_{\mathsf{c}}]} (x_{\mathsf{c}} - x)(x - x_{\mathsf{b}}) \quad = \quad C[\mathsf{a}, \mathsf{d}] + C[\mathsf{b}, \mathsf{c}].
\end{aligned}$$

Here, the inequality follows from equations (1)-(3). $\qquad\square$

Next, let us implicitly define a matrix $A \in \mathbb{R}^{\{1, \ldots, d\} \times \{1, \ldots, d\}}$ such that $A[k, j] = MSE[i - 1, k] + C[k, j]$. Importantly, $A$ *is not* stored in memory but admits constant time lookups as $MSE[i - 1, \cdot]$ is stored and $C$ is efficiently computable (Section 3). Also, $C$ satisfies the quadrangle inequality and thus $A$ is a totally monotone matrix [36], i.e., for any $\mathsf{a} < \mathsf{b}$ and $\mathsf{c} < \mathsf{d}$: $(A[\mathsf{a}, \mathsf{c}] > A[\mathsf{b}, \mathsf{c}]) \implies (A[\mathsf{a}, \mathsf{d}] > A[\mathsf{b}, \mathsf{d}])$. By applying the SMAWK algorithm [37], which finds the row minimas of an implicitly defined totally monotone matrix, on $A^T$, we obtain in $O(d)$ time and space the indices $k_j = \operatorname{argmin}_{k \in \{1, \ldots, d\}} A[k, j]$ for all $j \in \{1, \ldots, d\}$. This immediately gives the next row of the dynamic program, as $MSE[i, j] = MSE[i - 1, k_j] + C[k_j, j]$.

The resulting solution, which we call QUIVER, is given in Algorithm 1 and requires just $O(s \cdot d)$ time and space to compute the optimal quantization values.

## 5 The Accelerated QUIVER Algorithm

To accelerate QUIVER, we rely on the observation that while the problem is non-convex for $s > 3$, it admits a closed-form solution when $s = 3$.

Denoting by $C^2[k, j] = \min_{b \in \{k, \ldots, j\}} (C[k, b] + C[b, j])$ the optimal MSE of quantizing the range $[x_k, x_j]$ using three quantization values (at $x_k, x_b, x_j$), we show how to compute $C^2$ in constant time.

Namely, consider adding a quantization value $q \in [x_k, x_j]$ (not necessarily in $X$) between two existing quantization values $x_k$ and $x_j$. Let us define the sum of variances of all input entries in $[x_k, x_j]$ as a function of $q$: $Q(q) = \sum_{x \in [x_k, q]} (q - x)(x - x_k) + \sum_{x \in (q, x_j]} (x_j - x)(x - q)$. This function is differentiable in $[x_k, x_j] \setminus X$, and we get: $\frac{dQ(q)}{dq} = \sum_{x \in [x_k, q]} (x - x_k) - \sum_{x \in (q, x_j]} (x_j - x)$.

---
**Algorithm 1** QUIVER
---

1: **Input:** $X \in \mathbb{R}^d, s \in \mathbb{N}$.          ▷ $X$ is sorted.
2: Preprocess($X$)        ▷ Enables computing $C[k, j]$ in constant time (Section 3).
3: **for** $j = 2$ **to** $d$ **do**
4:     $MSE[2, j] = C[1, j]$
5: **for** $i = 3$ **to** $s$ **do**
6:     $K[i, \cdot] = $ SMAWK($A$)        ▷ Where $A[k, j] \triangleq MSE[i - 1, k] + C[k, j]$    $\forall k, j$.
7:     $MSE[i, j] = MSE[i - 1, K[i, j]] + C[K[i, j], j]$ for all $j \in \{i, \dots, d\}$.
8: $Q = \{x_1, x_d\}$
9: $j = d$
10: **for** $i = s$ **down to** $3$ **do**
11:     $j = K[i, j]$
12:     $Q = Q \cup \{x_j\}$
13: **return** $Q$

---
**Algorithm 2** Accelerated QUIVER
---

1: **Input:** $X \in \mathbb{R}^d, s \in \mathbb{N}$.          ▷ $X$ is sorted.
2: Preprocess($X$)        ▷ Enables computing $C[k, j]$ and $C^2[k, j]$ in constant time.
3: $s' = (s \mod 2)$
4: **if** $s' = 0$ **then**
5:     **for** $j = 2$ **to** $d$ **do**
6:        $MSE[2, j] = C[1, j]$
7: **else**
8:     **for** $j = 3$ **to** $d$ **do**
9:        $MSE[3, j] = C^2[1, j]$
10: **for** $i = 2$ **to** $\lfloor s/2 \rfloor$ **do**
11:     $K[i, \cdot] = $ SMAWK($B$)       ▷ Where $B[k, j] \triangleq MSE[2 \cdot (i - 1) + s', k] + C^2[k, j]$    $\forall k, j$.
12:     $MSE[2 \cdot i + s', j] = MSE[2 \cdot (i - 1) + s', K[i, j]] + C^2[K[i, j], j]$     $\forall j \in \{i, \dots, d\}$.
13: $Q = \{x_1, x_d\}$
14: $j = d$
15: **for** $i = \lfloor s/2 \rfloor$ **down to** $2$ **do**
16:     $b^* = \text{argmin}_{b \in \{K[i,j], \dots, j\}} \left( C[K[i, j], b] + C[b, j] \right)$       ▷ Takes $O(1)$ time.
17:     $j = K[i, j]$
18:     $Q = Q \cup \{x_j, x_{b^*}\}$
19: **if** $s' = 1$ **then**
20:     $b^* = \text{argmin}_{b \in \{0, \dots, j\}} \left( C[0, b] + C[b, j] \right)$       ▷ Takes $O(1)$ time.
21:     $Q = Q \cup \{x_{b^*}\}$
22: **return** $Q$

---

Notice that the derivative is monotonically non-decreasing and for any $\ell \in \{k, k+1, \dots, j-1\}$ the derivative is fixed (independent of $q$) over any interval $(x_\ell, x_{\ell+1})$. This means that $Q(q)$ is minimized at $u = \inf_q(\frac{dQ(q)}{dq} \geq 0)$, where $u \in X$. Denote by $b^*_{k,j} \in \{k, \dots, j\}$ the value such that $x_{b^*_{k,j}} = u$. Notice that while $\frac{dQ(u)}{dq}$ may not be defined, we have that $\lim_{h \to 0^+} \frac{dQ(u+h)}{dq} \geq 0$ is well-defined.

We thus require $\sum_{i=k+1}^{b^*_{k,j}} (x_i - x_k) - \sum_{i=b^*_{k,j}+1}^{j} (x_j - x_i) \geq 0$. With some simplifications, this is equivalent to: $\sum_{i=k+1}^{j} x_i - (b^*_{k,j} - k)x_k - (j - b^*_{k,j})x_j \geq 0$, yielding $b^*_{k,j} \geq \frac{jx_j - kx_k - \sum_{i=k+1}^{j} x_i}{x_j - x_k}$.

As $b^*_{k,j}$ is an integer, we get a formula for $C^2[k, j]$ that can be computed in constant time using: $b^*_{k,j} = \lceil \frac{jx_j - kx_k - \sum_{i=k+1}^{j} x_i}{x_j - x_k} \rceil = \lceil \frac{jx_j - kx_k - (\beta_j - \beta_k)}{x_j - x_k} \rceil$. That is, for any $1 \leq k \leq j \leq d$ we have that $C^2[k, j] = C[k, b^*_{k,j}] + C[b^*_{k,j}, j]$ is the sum of the variances in quantizing the entries in $[x_k, x_j]$ using the quantization values $\{x_k, x_{b^*_{k,j}}, x_j\}$.

We can then use this method to halve the required number of invocations of SMAWK by always using it to pick the *second-next* quantization value and computing the optimal quantization value in between directly. Our accelerated dynamic program is then given by:

$$MSE[i,j] = \begin{cases} \min\limits_{k \in \{i,\dots,j\}} \quad MSE[i-2,k] + C^2[k,j] & i > 3 \\ C^2[1,j] & i = 3 \\ C[1,j] & i = 2 \end{cases},$$

and the resulting pseudo-code for Accelerated QUIVER is given by Algorithm 2. Similarly to QUIVER, we start by initializing the first row of $MSE$. Importantly, we now separate the even $s$ case (lines 5-6), in which we initialize the row using $C$, and the odd case, where we use $C^2$ (lines 8-9). That is, the odd $s$ case 'skips' a quantization value that we later determine separately (lines 19-21). Next, denoting $s' = (s \mod 2)$, we proceed with $\lfloor s/2 \rfloor - 1$ invocations of the SMAWK algorithm (lines 10-12), applied on the implicitly defined matrix $B[k,j] \triangleq MSE[2 \cdot (i-1) + s', K[i,j]] + C^2[K[i,j],j]$. The output yields the minimizers of $MSE[2 \cdot i + s', j]$ used for reconstruction. In the reconstruction step (lines 15-21), we fill in the missing quantization values by finding the optimal value between every two outputs from the dynamic program minimizers $K$.

Overall, the Accelerated QUIVER algorithm requires at most half of the number of SMAWK invocations compared to QUIVER and at most half of the memory to store $K$ and $MSE$.

To establish correctness, we state the following lemma, whose proof appears in Appendix C.

**Lemma 5.1.** $C^2$ *satisfies the quadrangle inequality.*

In Appendix D, we discuss why this approach is not suitable for further acceleration by placing more than one quantization value in $[x_a, x_c]$.

## 6    The Approximate QUIVER Algorithm

We now show how the usage of *quantization value discretization* gives a controllable tradeoff between accuracy and speed. Intuitively, by allowing the quantization values to be placed only on a uniform grid of controllable size $m + 1 \geq s$ (for some $m \in \mathbb{N}^+$), we can accelerate the computation at the cost of a small additional error. Importantly, while the quantization values are from a discretized set of possibilities, we compute the *optimal* subset of discretized values for the *original input vector*.

To that end, consider the discrete set $S = \left\{ x_1 + \ell \cdot \frac{x_d - x_1}{m} \mid \ell \in \{0, \dots, m\} \right\}$ . Our goal is then to find $Q \in \binom{S}{s}$ that minimizes the sum of variances for the original input. Denoting $s_\ell = x_1 + \ell \cdot \frac{x_d - x_1}{m}$, we modify our preprocessing scheme to consider the discretization:

$$\alpha_\ell = \sum_{x \in [s_0, s_\ell]} 1 \quad , \quad \beta_\ell = \sum_{x \in [s_0, s_\ell]} x \quad , \quad \gamma_\ell = \sum_{x \in [s_0, s_\ell]} x^2 \qquad \forall \ell \in \{1, \dots, m\} \ .$$

As we explain in Appendix E, we can compute these values in $O(d)$ time and space.

Using these arrays, we can express the sum of variances of all input entries between two quantization values $s_k, s_j$ as follows:

$$\begin{aligned} C_m[k,j] &= \sum_{x \in [s_k, s_j]} (s_j - x)(x - s_k) = \sum_{x \in (s_k, s_j]} (s_j - x)(x - s_k) \\ &= -s_j \cdot s_k \cdot \sum_{x \in (s_k, s_j]} 1 + (s_j + s_k) \cdot \sum_{x \in (s_k, s_j]} x - \sum_{x \in (s_k, s_j]} x^2 \\ &= -s_j \cdot s_k \cdot (\alpha_j - \alpha_k) + (s_j + s_k) \cdot (\beta_j - \beta_k) - (\gamma_j - \gamma_k). \end{aligned}$$

Note that the quadrangle inequality trivially holds for this extension. The resulting algorithm, termed Approximate QUIVER (or in short, Apx. QUIVER), proceeds as QUIVER with $C_m$ instead of $C$, except for the reconstruction stage where we pick $Q$ from $S$ instead of the input $X$. Apx. QUIVER, whose pseudo-code is given in Appendix F, runs in space and time complexities of $O(d + m \cdot s)$.

We next analyze the approximation guarantee of Apx. QUIVER. Denote by $\texttt{opt}_{X,s}$ the optimal MSE attainable for $X$ using $s$ quantization values, and by $\texttt{AQ}_{X,2s-2}$ the MSE of Apx. QUIVER with $2s-2$ values. We prove that the MSE of Apx. QUIVER with $2s-2$ quantization values is close to the optimal algorithm with $s$ values. In practice, we generally find Apx. QUIVER does better than the bound below, and for moderate $m$, it is nearly optimal.

**Lemma 6.1.** *For any $X, s, m$ we have* $\texttt{AQ}_{X,2s-2} \leq \texttt{opt}_{X,s} + \frac{d \cdot (x_d - x_1)^2}{4m^2} \leq \texttt{opt}_{X,s} + \frac{d \cdot \|X\|_2^2}{2m^2}$.

*Proof.* Let $Q^* \subseteq X$ be the optimal solution with $|Q^*| \leq s$. For any $q \in Q^*$, denote by $\underline{q} = \max\{s_\ell \in S \mid s_\ell \leq q\}$ and $\overline{q} = \min\{s_\ell \in S \mid s_\ell \geq q\}$. Consider the solution $\widetilde{Q} = \{\underline{q}, \overline{q} \mid q \in Q^*\}$. Note that $|\widetilde{Q}| \leq 2s - 2$ as $x_1, x_d \in Q^*$ and $\overline{x_1} = \underline{x_1}$ and $\overline{x_d} = \underline{x_d}$. Also, $\widetilde{Q} \subseteq S$ and is thus a valid solution of Apx. QUIVER. Thus, $\texttt{AQ}_{X,2s-2}$ is upper bounded by the MSE when using $\widetilde{Q}$.

Next, consider $x \in X$ and let $a_x = \max\{q \in Q^* \mid q \leq x\}$ and $b_x = \min\{q \in Q^* \mid q \geq x\}$ be the values between which $x$ is stochastically quantized in $Q^*$. We consider two cases:

- $x \in [\underline{a_x}, \overline{a_x}] \cup (\underline{b_x}, \overline{b_x}]$. In this case, when using $\widetilde{Q}$, we have that $x$ is quantized in an interval of size $(x_d - x_1)/m$ and thus its variance is bounded by $(x_d - x_1)^2/4m^2$.

- $x \in [\overline{a_x}, \underline{b_x}]$, in this case, using $\widetilde{Q}$, $x$ is quantized between $\overline{a_x}$ and $\underline{b_x}$, yielding a variance of $(\underline{b_x} - x)(x - \overline{a_x}) \leq (b_x - x)(x - a_x)$, i.e., lower than the variance under $Q^*$.

As the two cases capture all options, summing the variances over all $x \in X$ yields the result. □

In terms of the *vector normalized MSE* (vNMSE),[1] which is a normalized MSE measure given by $\frac{\mathbb{E}\left[\|X - \widehat{X}\|_2^2\right]}{\|X\|_2^2}$, Apx. QUIVER with $2s-2$ quantization values achieves an additive $\frac{d}{2m^2}$ term to the optimal vNMSE when using $s$ quantization values.

However, the first inequality of Lemma 6.1 is generally much tighter than the second that uses the squared norm. For example, if the entries of $X$ were i.i.d. $U[a, b]$ random variables, for some constants $a < b$ then $(x_d - x_1)^2 = O(1)$ while $\|X\|_2^2 = \Theta(d)$. Similarly, for i.i.d $\mathcal{N}(\mu, \sigma^2)$ entries for constants $\mu, \sigma$ we have $(x_d - x_1)^2 = O(\log d)$ while $\|X\|_2^2 = \Theta(d)$ (both with high probability).

## 7    Evaluation

We evaluate our algorithms' empirical vNMSE and runtime against SOTA ASQ solutions.

**Setup.**    We implement all algorithms in C++. Unless stated otherwise, we use a `g4dn.4xlarge` AWS EC2 server with custom Intel Cascade Lake CPUs with 64 GB RAM and Ubuntu 22.04 OS and average all results over 5 seeds.

**Acceleration Speedup**    Appendix G shows the speedup attainable by Accelerated QUIVER. As we show, Accelerated QUIVER is consistently faster than QUIVER, providing up to $5.4\times$ speedup.

**Distributions.**    All experiments are done with vectors whose entries are independent and identically distributed. We present results for the LogNormal distribution and defer to Appendix H results for Normal, Exponential, TruncNorm, and Weibull distributions. As mentioned, these distributions are of interest as they are reported to capture gradients, model weights and activations (see Section 1).

---

[1]This metric is standard in quantization works (e.g., see [17] and the references therein). It enables us to reason about the results among different dimensions and distributions.

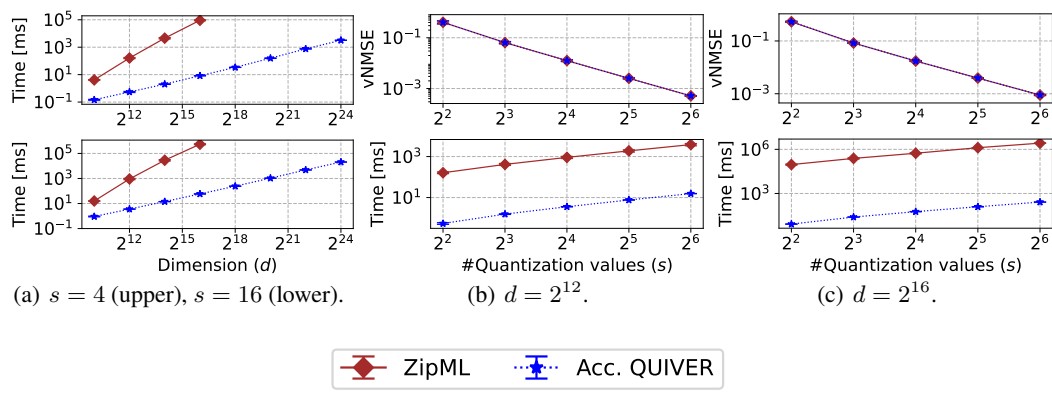

(a) $s = 4$ (upper), $s = 16$ (lower).  (b) $d = 2^{12}$.  (c) $d = 2^{16}$.

ZipML    Acc. QUIVER

Figure 2: Comparing exact solutions with LogNormal$(0, 1)$ distributed input.

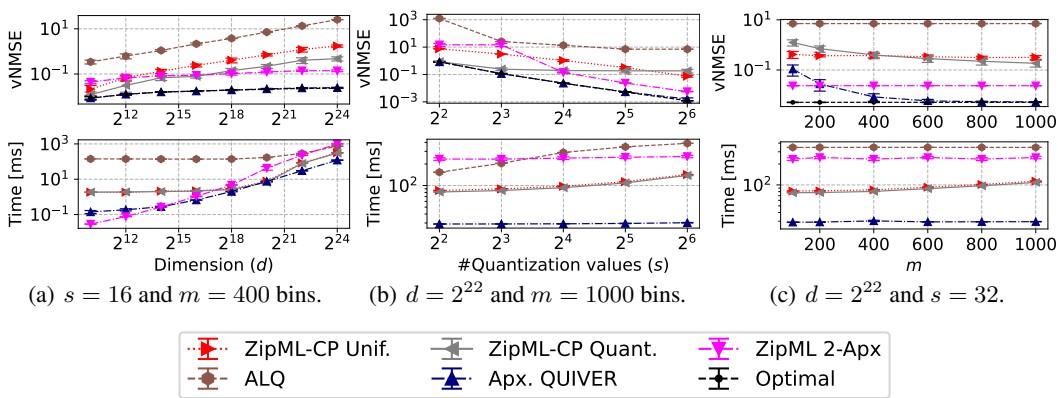

(a) $s = 16$ and $m = 400$ bins.  (b) $d = 2^{22}$ and $m = 1000$ bins.  (c) $d = 2^{22}$ and $s = 32$.

ZipML-CP Unif.    ZipML-CP Quant.    ZipML 2-Apx
ALQ    Apx. QUIVER    Optimal

Figure 3: Comparing approximate solutions with LogNormal$(0, 1)$ distributed input.

**Baselines.** We evaluate Accelerated QUIVER and compare its runtime to ZipML [26]. For the approximate variants, we evaluate Apx. QUIVER and compare it with three approximation variants of ZipML proposed in [26], namely ZipML-CP Quantiles, ZipML-CP Uniform, and ZipML 2-Approximation. ZipML-CP is an algorithm that runs the exact ZipML algorithm on a subset of the points called 'Candidate Points'. Since ZipML runs in $O(d^2 s)$ time, here we use M candidate points to get $O(d + M^2 s)$ time. ZipML 2-Apx is an algorithm that computes an approximate solution in $O(d \log d + s^3)$ time. It guarantees that its sum of variances is at most twice that of an optimal solution with $\lfloor s/2 \rfloor$ quantization values. We also compare with the recently proposed ALQ [28], which is an algorithm that finds good quantization values for a truncated normal distribution. It samples several gradients (by computing the gradient of several random batches) to fit the truncated normal parameters. To be fair to ALQ, since we evaluate a single-shot quantization scenario, we calculate the exact mean, variance, and support parameters for the input vector. This then runs for several (we used 10, as in their released code) iterations, so in total, they compute $\approx 10s$ integrals. While theoretically requiring $O(d)$ time, in a model where such integral calculation takes constant time, this is markedly slower than other approaches. We note that it is possible that with low-precision integral calculations, one may improve the runtime, but the error (which is already not competitive) will degrade further. We further discuss these approximation algorithms in Appendix I.

**Exact algorithms experiments.** The results are presented in Figure 2. Figure 2(a) shows the runtime for optimally solving the ASQ problem for different dimensions and $s$. As shown, all our solutions are markedly faster than ZipML, which we are unable to run for dimensions $d \geq 2^{17}$ due to its prohibitively large memory requirements. The asymptotic difference ($O(s \cdot d^2)$ for ZipML and $O(s \cdot d)$ for Accelerated QUIVER) is clearly visible in the different slopes on the log-log plot. As

a result, Accelerated QUIVER can efficiently quantize vectors. For example, Acc. QUIVER can compute the optimal 4-bit ($s = 16$) quantization values for a 1M-sized vector in under a second.

Next, Figure 2(b) and Figure 2(c) show the vNMSE and runtime with respect to the number of quantization values $s$ for $d = 2^{12}$ and $d = 2^{16}$. As shown, the vNMSE decays linearly with $s$ while the runtime increases linearly. Even for these small dimensions, our algorithms are orders of magnitude faster than ZipML.

**Approximate algorithms experiments.** The comparison results are presented in Figure 3. It is evident in Figure 3(a) that approximate solutions are significantly faster than exact ones. Also, Apx. QUIVER offers both near-optimal vNMSE and the fastest runtime as the dimension increases. As shown in Figures 3(b) and 3(c), Apx. QUIVER offers these advantages for different $s, m$ values.

Notably, on a commodity PC, Apx. QUIVER can compute near-optimal 4-bit quantization values ($s = 16$) for a vector with $d = 2^{20}$ entries in just six milliseconds, and about 70ms for $d = 2^{24}$, potentially enabling quantizing vectors on the fly for many applications.

# 8 Discussion

In this paper, we presented algorithms for the Adaptive Stochastic Quantization (ASQ) problem with improved space and time complexities compared to the state of the art. For parameters of interest, our exact algorithms are up to four orders of magnitude faster compared to the alternatives while using markedly less memory. To potentially enable on-the-fly adaptive quantization of vectors, we also introduce an approximate algorithm with strong guarantees that runs faster while being significantly more accurate than other approximate solutions.

**Limitations:** QUIVER is not GPU friendly, and it remains an interesting future work to design GPU-friendly ASQ algorithms. Also, similarly to previous works (e.g., [26]), our exact solution assumes that the input vector is sorted. Otherwise, the runtime is increased to $O(d \cdot \log d + s \cdot d)$. We note that Apx. QUIVER does not require the vector to be sorted and the time complexity remains $O(d + s \cdot m)$ even for non-sorted inputs, making it even more appealing compared to the exact solutions.

**Offloading Computation to a GPU:** For exact algorithms, one can sort the input vector on a GPU, bringing the CPU solution complexity to $O(s \cdot d)$ which is faster for large vectors. In practice, GPU sorting is rarely the bottleneck; indeed, in Appendix J we measure the time it takes to sort the vector on a T4 GPU, and also to quantize the vector after an ASQ outputs the optimal quantization values. For example, the sorting and quantization time for a $1M$-sized vector sums up to only 4ms where the runtime of Accelerated QUIVER is about one second.

**Generalizing the algorithms for weighted inputs:** An interesting generalization of the ASQ problem is the weighted variant, where each entry $x_i \in X$ is associated with a weight $w_i \in \mathbb{R}$ and the goal is to minimize the weighted sum of variances $\sum_{i=1}^{d} (x_i - \widehat{x_i})^2 \cdot w_i$. This variant is useful when, instead of getting an input vector, one wishes to solve ASQ for an empirical distribution. In Appendix K we explain how our algorithms and their analyses generalize to the weighted case, while maintaining the $O(d \cdot s)$ and $O(d + M \cdot s)$ runtime and space complexities for QUIVER and Apx. QUIVER accordingly. Our measurements indicate that the weighted variants are only 10-20% slower than their unweighted counterparts.

**Reproducability:** All our results are reproducible and our code is open sourced [30].

## Acknowledgments and Disclosure of Funding

We thank Wenchen Han for his insightful comments and suggestions. Michael Mitzenmacher was supported in part by NSF grants CCF-2101140, CNS-2107078, and DMS-2023528.

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

---

**Algorithm 3** Basic Dynamic Programming Algorithm

---

1: **Input:** $X \in \mathbb{R}^d, s \in \mathbb{N}$.
2: Compute $C : [d] \times [d] \to \mathbb{R}^+$ using $X$.
3: **for** $j = 2$ **to** $d$ **do**
4:      $MSE[2, j] = C[1, j]$
5: **for** $i = 3$ **to** $s$ **do**
6:      **for** $j = i$ **to** $d$ **do**
7:          $MSE[i, j] = \min_{k \in \{i,\ldots,j\}} MSE[i-1, k] + C[k, j]$
8: $j = d$
9: $Q = \{x_1, x_d\}$
10: **for** $i = s$ **down to** $3$ **do**
11:      $j = \mathrm{argmin}_{k \in \{i,\ldots,j\}} \quad MSE[i-1, k] + C[k, j]$
12:      $Q = Q \cup \{x_j\}$
13: **return** $Q$

---

## A    Basic Algorithm

We now describe a simple algorithm that finds the optimal quantization values using the dynamic program, with pseudo-code given by Algorithm 3. After initialization (lines 2-4), the algorithm iteratively computes $MSE[i, \cdot]$ given $MSE[i-1, \cdot]$ (lines 5-7) and traces back the optimal quantization values given the solution (lines 8-12).

## B    The SMAWK Algorithm [37]

Here, we provide some intuition into how SMAWK operates and achieves its efficiency. The SMAWK algorithm has four main steps:

- **Pruning Phase:** Remove columns that cannot possibly contain a row maximum. This is done by comparing each column with its neighbors and discarding those that cannot be maxima based on the totally monotone property. At the end of this phase, the number of columns can be no larger than the number of rows.

- **Recursive Reduction:** The algorithm reduces the problem size by considering a subset of the rows and columns. It selects every other row and recursively solves the reduced problem.

- **Candidate Set:** After solving the smaller problem, the solution provides candidate columns for the original problem. The algorithm only needs to consider these columns to find the maxima for the skipped rows.

- **Merge Phase:** Combine the results from the reduced problem with the candidate set to find the maximum for each original row.

Regarding efficiency, the SMAWK algorithm achieves a time complexity of $O(d)$ for a $d \times d$ matrix. This efficiency is due to the recursive reduction of the problem size and the properties of totally monotone matrices that limit the number of comparisons needed. Namely, the pruning step takes $O(\#cols)$, where $\#cols$ is the number of columns still being considered. The crux is that the recursive step happens after the pruning, which means that the recursive invocation happens with a number of columns that is, at most, double the number of rows (as the number of rows is halved). This means that the overall complexity of each recursive step is proportional to the number of rows, yielding the recursion: $T(n) = T(n/2) + O(n) = O(n)$. A simple example Python implementation (by David Eppstein) appears here [38]. Our implementation is in optimized C++ [30].

## C    Proof of Lemma 5.1

**Lemma 5.1.** $C^2$ *satisfies the quadrangle inequality.*

*Proof.* The lemma claims that, for any $\mathsf{a} \le \mathsf{b} \le \mathsf{c} \le \mathsf{d}$:
$$C^2[\mathsf{a}, \mathsf{c}] + C^2[\mathsf{b}, \mathsf{d}] \le C^2[\mathsf{a}, \mathsf{d}] + C^2[\mathsf{b}, \mathsf{c}].$$

Recall that for any $a \le c \in \{1, \ldots, d\}$, we denote
$$b^*_{a,c} = \operatorname*{argmin}_{b \in \{a, \ldots, c\}} C[a, b] + C[b, c].$$

We prove the lemma by a case analysis:

- Case $b^*_{\mathsf{b},\mathsf{c}} \le b^*_{\mathsf{a},\mathsf{d}}$. In this case, we have that:
$$
\begin{aligned}
C^2(\mathsf{a}, \mathsf{c}) + C^2(\mathsf{b}, \mathsf{d}) &= C(\mathsf{a}, b^*_{\mathsf{a},\mathsf{c}}) + C(b^*_{\mathsf{a},\mathsf{c}}, \mathsf{c}) + C(\mathsf{b}, b^*_{\mathsf{b},\mathsf{d}}) + C(b^*_{\mathsf{b},\mathsf{d}}, \mathsf{d}) \\
&\underset{(i)}{\le} C(\mathsf{a}, b^*_{\mathsf{b},\mathsf{c}}) + C(b^*_{\mathsf{b},\mathsf{c}}, \mathsf{c}) + C(\mathsf{b}, b^*_{\mathsf{a},\mathsf{d}}) + C(b^*_{\mathsf{a},\mathsf{d}}, \mathsf{d}) \\
&\underset{(ii)}{\le} C(\mathsf{b}, b^*_{\mathsf{b},\mathsf{c}}) + C(b^*_{\mathsf{b},\mathsf{c}}, \mathsf{c}) + C(\mathsf{a}, b^*_{\mathsf{a},\mathsf{d}}) + C(b^*_{\mathsf{a},\mathsf{d}}, \mathsf{d}) \\
&= C^2(\mathsf{b}, \mathsf{c}) + C^2(\mathsf{a}, \mathsf{d}).
\end{aligned}
$$
  Here, the Inequality $(i)$ follows from the definition of $b^*_{a,c}$ that minimizes the MSE over the interval $[x_\mathsf{a}, x_\mathsf{c}]$ and $b^*_{b,d}$ that minimizes it over $[x_\mathsf{b}, x_\mathsf{d}]$. Inequality $(ii)$ follows from the quadrangle inequality of $C$ (Lemma 4.2), as $\mathsf{a} \le \mathsf{b} \le b^*_{\mathsf{b},\mathsf{c}} \le b^*_{\mathsf{a},\mathsf{d}}$, and thus
$$C(\mathsf{a}, b^*_{\mathsf{b},\mathsf{c}}) + C(\mathsf{b}, b^*_{\mathsf{a},\mathsf{d}}) \le C(\mathsf{b}, b^*_{\mathsf{b},\mathsf{c}}) + C(\mathsf{a}, b^*_{\mathsf{a},\mathsf{d}}).$$

- Case $b^*_{\mathsf{b},\mathsf{c}} > b^*_{\mathsf{a},\mathsf{d}}$. In this case, we have that:
$$
\begin{aligned}
C^2(\mathsf{a}, \mathsf{c}) + C^2(\mathsf{b}, \mathsf{d}) &= C(\mathsf{a}, b^*_{\mathsf{a},\mathsf{c}}) + C(b^*_{\mathsf{a},\mathsf{c}}, \mathsf{c}) + C(\mathsf{b}, b^*_{\mathsf{b},\mathsf{d}}) + C(b^*_{\mathsf{b},\mathsf{d}}, \mathsf{d}) \\
&\underset{(i)}{\le} C(\mathsf{a}, b^*_{\mathsf{a},\mathsf{d}}) + C(b^*_{\mathsf{a},\mathsf{d}}, \mathsf{c}) + C(\mathsf{b}, b^*_{\mathsf{b},\mathsf{c}}) + C(b^*_{\mathsf{b},\mathsf{c}}, \mathsf{d}) \\
&\underset{(ii)}{\le} C(\mathsf{b}, b^*_{\mathsf{b},\mathsf{c}}) + C(b^*_{\mathsf{b},\mathsf{c}}, \mathsf{c}) + C(\mathsf{a}, b^*_{\mathsf{a},\mathsf{d}}) + C(b^*_{\mathsf{a},\mathsf{d}}, \mathsf{d}) \\
&= C^2(\mathsf{b}, \mathsf{c}) + C^2(\mathsf{a}, \mathsf{d}).
\end{aligned}
$$
  Here, the Inequality $(i)$ follows again from $b^*_{a,c}$ and $b^*_{b,d}$ being optimal for $[x_\mathsf{a}, x_\mathsf{c}]$ and $[x_\mathsf{b}, x_\mathsf{d}]$. Inequality $(ii)$ follows from the quadrangle inequality of $C$, as $b^*_{\mathsf{a},\mathsf{d}} \le b^*_{\mathsf{b},\mathsf{c}} \le \mathsf{c} \le \mathsf{d}$ and, therefore,
$$C(b^*_{\mathsf{a},\mathsf{d}}, \mathsf{c}) + C(b^*_{\mathsf{b},\mathsf{c}}, \mathsf{d}) \le C(b^*_{\mathsf{a},\mathsf{d}}, \mathsf{d}) + C(b^*_{\mathsf{b},\mathsf{c}}, \mathsf{c}).$$

Together, this concludes the proof. $\qquad \square$

## D No apparent closed-form solution for $s > 3$

We explain why our acceleration method from Section 4 fails for $s > 3$. Consider computing the location of two additional quantization values $b \le u$ between $x_a$ and $x_c$.

Similarly to the above analysis, we define by $Q(b, u)$ the resulting sum of variances for all entries in $[x_a, x_c]$. Then:

$$Q(b, u) = \sum_{x \in [x_a, b]} (b - x)(x - x_a) + \sum_{x \in (b, u]} (u - x)(x - b) + \sum_{x \in (u, x_c]} (x_c - x)(x - u).$$

Computing the partial derivatives, we then get:
$$\frac{\partial Q(b, u)}{\partial b} = \sum_{x \in [x_a, b]} (x - x_a) - \sum_{x \in (b, u]} (u - x).$$
$$\frac{\partial Q(b, u)}{\partial u} = \sum_{x \in (b, u]} (x - b) - \sum_{x \in (u, x_c]} (x_c - x).$$

The challenge now is that both derivatives are non-continuous, and there are multiple indices $i, j$ such that $Q(x_i, x_j) < 0$ but $Q(x_{i+1}, x_j) \ge 0$ or $Q(x_i, x_{j+1}) \ge 0$. Accordingly, it seems unlikely that a closed-form solution that is computable in constant time follows from this approach.

---

**Algorithm 4** Apx. QUIVER

---

1: **Input:** $X \in \mathbb{R}^d, s, m \in \mathbb{N}$.
2: $S = \left\{ x_1 + \ell \cdot \frac{x_d - x_1}{m} \mid \ell \in \{0, \ldots, m\} \right\}$
3: $\texttt{Preprocess}(X, m)$         $\triangleright$ Enables computing $C_m[k, j]$ in constant time (Appendix E).
4: **for** $j = 2$ **to** $m$ **do**
5:     $MSE[2, j] = C_m[1, j]$
6: **for** $i = 3$ **to** $s$ **do**
7:     $K[i, \cdot] = \texttt{SMAWK}(Z)$            $\triangleright$ Where $Z[k, j] \triangleq MSE[i-1, k] + C_m[k, j]$    $\forall k, j$.
8:     $MSE[i, j] = MSE[i-1, K[i, j]] + C_m[K[i, j], j]$ for all $j \in \{i, \ldots, m\}$.
9: $Q = \{s_0, s_m\}$
10: $j = m$
11: **for** $i = s$ **down to** $3$ **do**
12:     $j = K[i, j]$
13:     $Q = Q \cup \{s_j\}$
14: **return** $Q$

---

## E   Preprosessing for Apx. QUIVER

Recall that, for $S = \left\{ x_1 + \ell \cdot \frac{x_d - x_1}{m} \mid \ell \in \{0, \ldots, m\} \right\}$ and $s_\ell = x_1 + \ell \cdot \frac{x_d - x_1}{m}$, our goal is to compute the following arrays in $O(d)$ time:

$$\alpha_\ell = \sum_{x \in [s_0, s_\ell]} 1 \quad , \quad \beta_\ell = \sum_{x \in [s_0, s_\ell]} x \quad , \quad \gamma_\ell = \sum_{x \in [s_0, s_\ell]} x^2 \qquad \forall \ell \in \{1, \ldots, m\} \ .$$

Denoting $\delta = \frac{x_d - x_1}{m}$, the first step is to make a pass over the input and for each $x \in X$ calculate $\ell_x = \left\lfloor \frac{x - x_1}{\delta} \right\rfloor$ and set

$$A_\ell = \sum_{x \mid \ell_x = \ell} 1 \quad , \quad B_\ell = \sum_{x \mid \ell_x = \ell} x \quad , \quad \Gamma_\ell = \sum_{x \mid \ell_x = \ell} x^2 \qquad \forall \ell \in \{1, \ldots, m\} \ .$$

Next, we make an $O(m)$ time pass to compute the cumulative sums:

$$\alpha_\ell = \sum_{i=1}^{\ell} A_i \quad , \quad \beta_\ell = \sum_{i=1}^{\ell} B_i \quad , \quad \gamma_\ell = \sum_{i=1}^{\ell} \Gamma_i \qquad \forall \ell \in \{1, \ldots, m\} \ .$$

We note that an optimization that proved useful for improving the runtime in practice is to remove empty intervals after the first step. That is, we retain only intervals for which $A_\ell > 0$, thus reducing the number of intervals from $m$ to $m' \le m$, which can be markedly smaller in practice.

## F   Apx. QUIVER Pseudo-code

We describe the pseudo-code of Apx. QUIVER, which is given by Algorithm 4. We start by preprocessing the input to obtain the $\alpha, \beta, \gamma$ arrays ( Line 3). Next, we initialize the first row of the matrix, which only has $m$ columns, using $C_m$ (Line 4). Follows are $s - 2$ invocations of the SMAWK algorithm, each yielding the next row in $MSE$ and its minimizers $K[i, \cdot]$ (Line 6). Finally, we compute the resulting quantization value set $Q$ from $K$ and $S$ (Line 11).

## G   QUIVER Acceleration Evaluation

Here, we evaluate by how much Accelerated QUIVER is faster than QUIVER. The results, depicted in Figure 4, show that Accelerated QUIVER is up to $5.4\times$ faster for $s = 3$ and is consistently faster throughout. Interestingly, the speedup is more significant in odd values of $s$. This is because the number of SMAWK invocations is $\lfloor s/2 \rfloor - 1$ in Accelerated QUIVER (e.g., it does not invoke SMAWK at all for $s = 3$, only once for $s = 5$, etc.), compared to $s - 2$ invocations in QUIVER.

# H  Additional evaluation results

**Additional evaluation results of exact solutions.**  We provide results for additional input vectors distributions: Normal (Figure 5), Exponential (Figure 6), Truncated Normal (Figure 7), and Weibull (Figure 8). As shown, all follow the same trends in terms of vNMSE, while the runtime is largely independent of the input distribution.

# I  ASQ Approximation Baselines

In the ZipML paper [26], the authors propose two heuristic methods for improving the runtime. The first heuristic includes calculating the optimal solution on a subset of $X$ called *candidate points* (CP); they further present an analysis that bounds the error with respect to the maximal difference between consecutive CPs and the maximal number of entries in $X$ between consecutive CPs; however, as they do not provide a way to select the CPs, we consider two natural choices: using Uniform CPs, i.e., $\left\{ x_1 + \ell \cdot \frac{x_d - x_1}{m} \mid \ell \in \{0, \ldots, m\} \right\}$.[2] This variant is termed 'ZipML-CP Unif.' in our evaluation. The second choice of CP is Quantiles, which uses the set $\left\{ x_{\lfloor 1 + \ell \cdot (d-1)/m \rfloor} \mid \ell \in \{0, \ldots, m\} \right\}$. This variant is termed 'ZipML-CP Quant.' in our evaluation.

The second heuristic has a bicretira MSE guarantee: using $2s$ quantization values, it ensures that the MSE is at most twice that of the optimal solution with $s$ quantization values. This variant is termed 'ZipML 2-Apx' in our evaluation.

---

[2]We note that this is different our histogram approach in two aspects: (i) we stochastically quantize $X$ into the set $S$ and (ii) we use weights to consider the number of entries in each histogram bin.

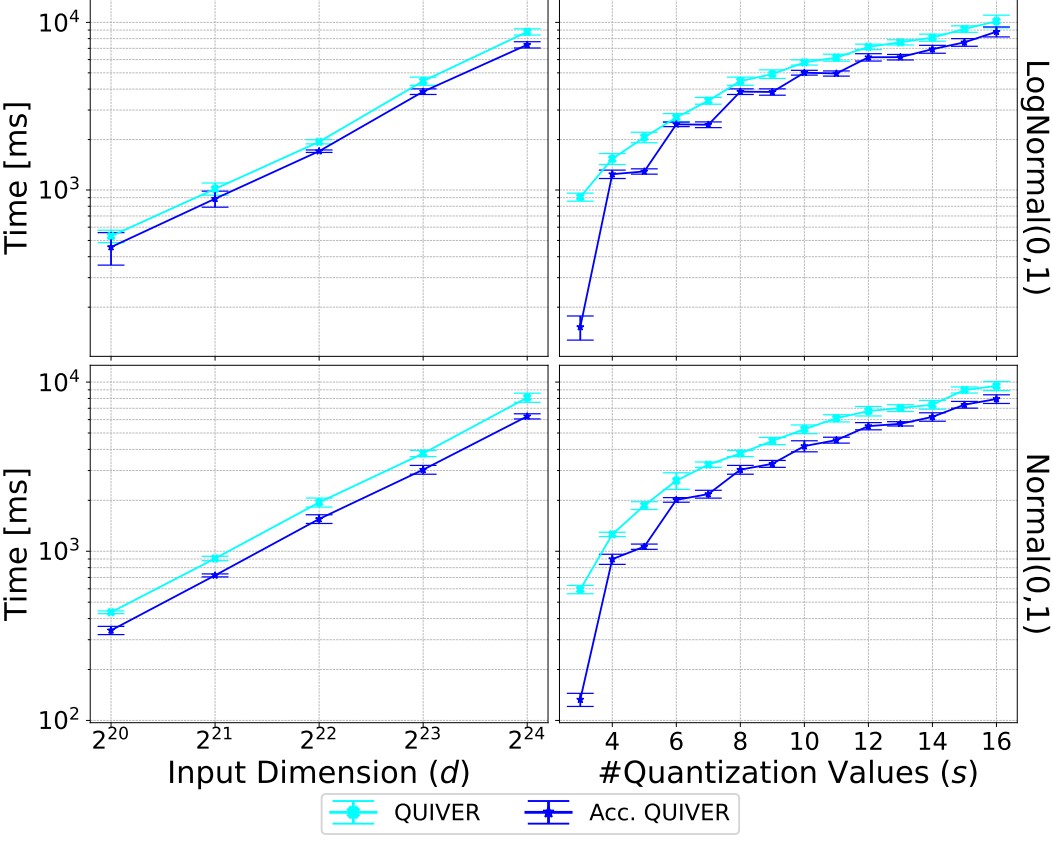

Figure 4: The speedup attainable by Accelerated QUIVER, as a function of $s$ (for fixed $d = 2^{23}$) and $d$ (for fixed $s = 8$), on the Normal and LogNormal distributions.

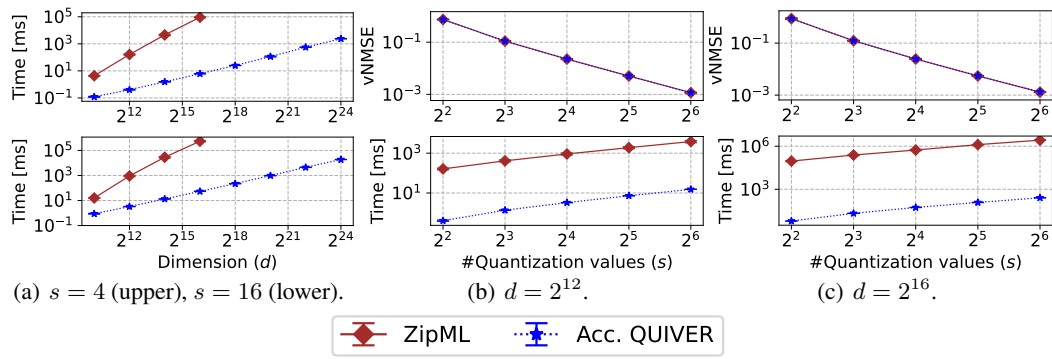

Figure 5: Comparing exact solutions with Normal$(0, 1)$ distributed input.

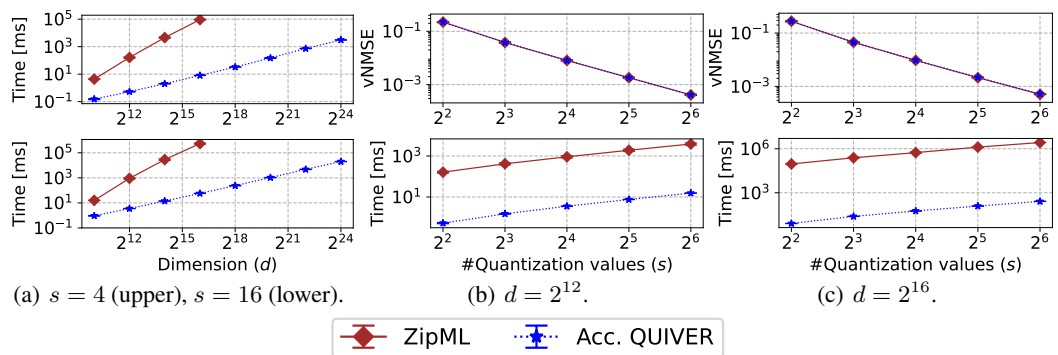

Figure 6: Comparing exact solutions with Exponential$(1)$ distributed input.

We also compare against ALQ [28], which fits the parameters of a truncated normal distribution to approximate the distribution of the input vector after normalizing it by its norm. It then uses an iterative solution to approximate the optimal quantization values of the fitted distribution up to the desired precision. As suggested by the authors, we use ten iterations, which were shown to converge to the optimal quantization values for the resulting (truncated normal) distribution.

**Additional evaluation results of approximate solutions.** Similarly, we show the approximation algorithms evaluation results for the various distributions and $s$ values: Normal (Figure 9), Exponential (Figure 10), Truncated Normal (Figure 11), and Weibull (Figure 12). Again, the runtime of all algorithms is weakly affected by the input distribution. Apx. QUIVER is always the most accurate for increasing $d$ values and has a near-optimal vNMSE when using a sufficient value for $m$ (e.g., $m \geq 400$) while being markedly faster than all alternatives.

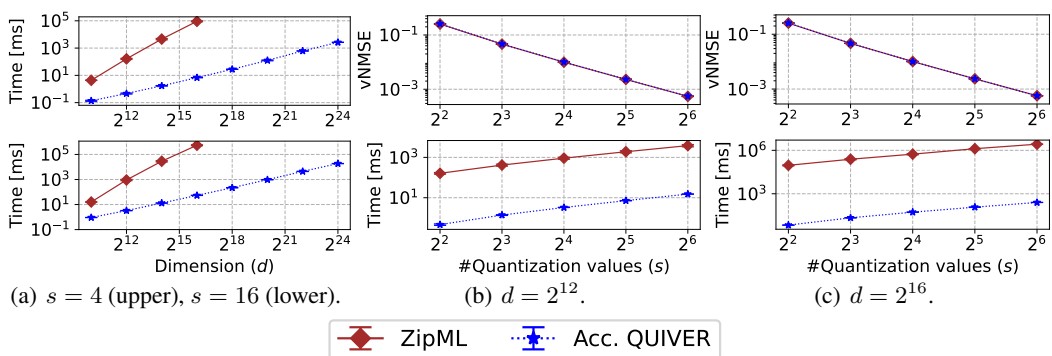

Figure 7: Exact solutions with TruncNorm$(\mu = 0, \sigma^2 = 1, a = -1, b = 1)$ distributed input.

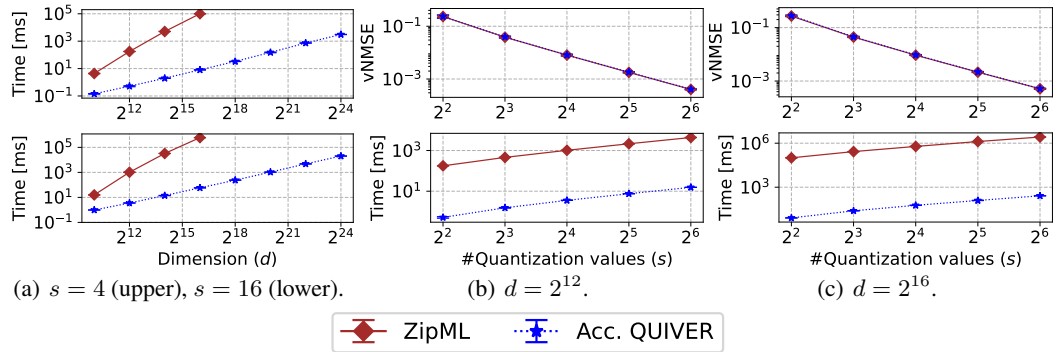

(a) $s = 4$ (upper), $s = 16$ (lower).  (b) $d = 2^{12}$.  (c) $d = 2^{16}$.

◆ ZipML  ⋆ Acc. QUIVER

Figure 8: Comparing exact solutions with Weibull$(1, 1)$ distributed input.

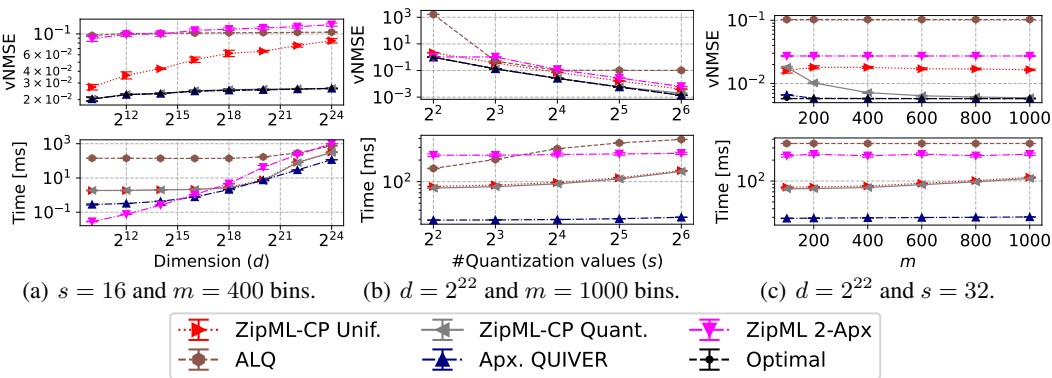

(a) $s = 16$ and $m = 400$ bins. (b) $d = 2^{22}$ and $m = 1000$ bins. (c) $d = 2^{22}$ and $s = 32$.



▶ ZipML-CP Unif.  ⋆ ZipML-CP Quant.  ▼ ZipML 2-Apx

● ALQ  ▲ Apx. QUIVER  ✦ Optimal



Figure 9: Comparing approximate solutions with Normal$(0, 1)$ distributed input.

## J  Additional Overheads

We measure the sort and quantize operations using the same EC2 server that is also equipped with an NVIDIA T4 GPU, PyTorch v2.1.2, and CUDA tool kit v12.3. As shown in Figure 13, both operations are fast even for large vectors, despite the usage of a somewhat weak GPU. This specific measurement was done over the LogNormal(0,1) distribution, but the sorting and quantization times are largely independent of the specific distribution and were similar to other tested distributions as well.

## K  Generalizing Our Algorithms to Weighted Inputs

We generalize our algorithms for processing sorted weighted inputs $X, W \in \mathbb{R}^d$ (where each entry has value $y_\ell$ and weight $w_\ell$ and $x_1 \le x_2 \le \ldots, x_d$).[3]

Most of the algorithmic parts only require a revised method for computing $C$ in constant time, which is achieved through the modified pre-processing procedure below.

For simplicity, we only discuss the basic QUIVER variant and leave the acceleration as future work.

**Pre-processing.** To allow constant time computation of weighted $C$, denoted $C_w$, for weighted inputs we need another auxiliary array. Namely, we define the following:

---

[3]Similarly to the unweighted case, the sorted vector requirement is only needed for the exact solutions.

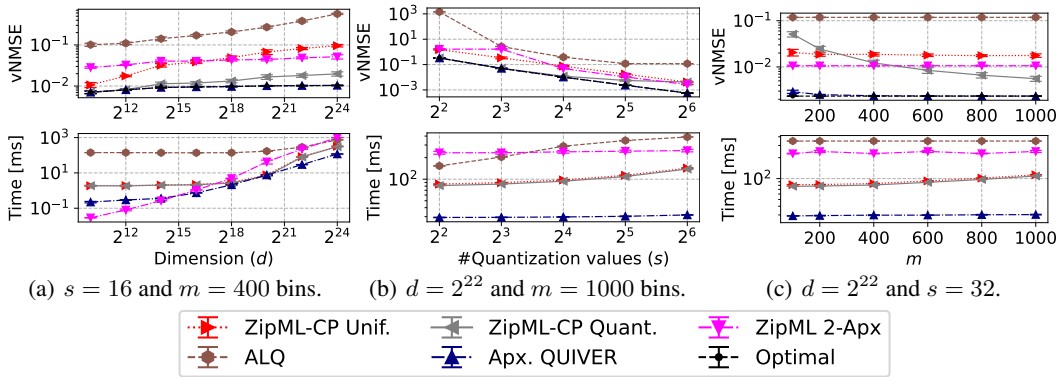

Figure 10: Comparing approximate solutions with Exponential(1) distributed input.

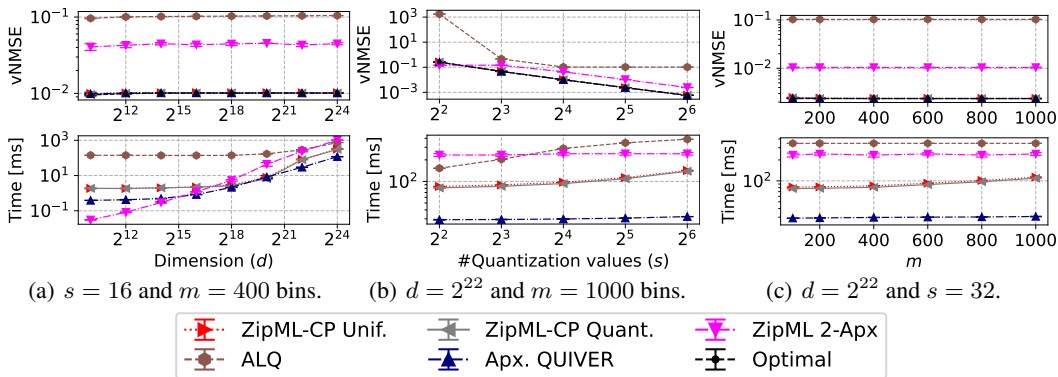

Figure 11: Approx. solutions with TruncNorm($\mu = 0, \sigma^2 = 1, a = -1, b = 1$) distributed input.

$$\alpha_j = \sum_{(x,w) \in X_j} w \qquad , \quad j \in \{1, \ldots, d\} \ ,$$

$$\beta_j = \sum_{(x,w) \in X_j} w \cdot x \quad , \quad j \in \{1, \ldots, d\} \ ,$$

$$\gamma_j = \sum_{(x,w) \in X_j} w \cdot x^2 \quad , \quad j \in \{1, \ldots, d\} \ .$$

Then, we can then write:

$$
\begin{aligned}
C_w[k, j] &= \sum_{x_\ell \in [x_k, x_j]} w \cdot (x_j - x_\ell)(x_\ell - x_k) \\
&= \sum_{x_\ell \in (x_k, x_j]} w \cdot (x_j - x_\ell)(x_\ell - x_k) \\
&= x_j \cdot x_k \cdot \sum_{x_\ell \in (x_k, x_j]} w_\ell + (x_j - x_k) \cdot \sum_{x_\ell \in (x_k, x_j]} w_\ell \cdot x_\ell - \sum_{x_\ell \in (x_k, x_j]} w_\ell \cdot x_\ell^2 \\
&= x_j \cdot x_k \cdot (\alpha_j - \alpha_k) + (x_j - x_k) \cdot (\beta_j - \beta_k) - (\gamma_j - \gamma_k).
\end{aligned}
$$

Observe that $C_w$ clearly satisfies the quadrangle inequality, and thus, the correctness follows. The approximation variant also follows similarly.

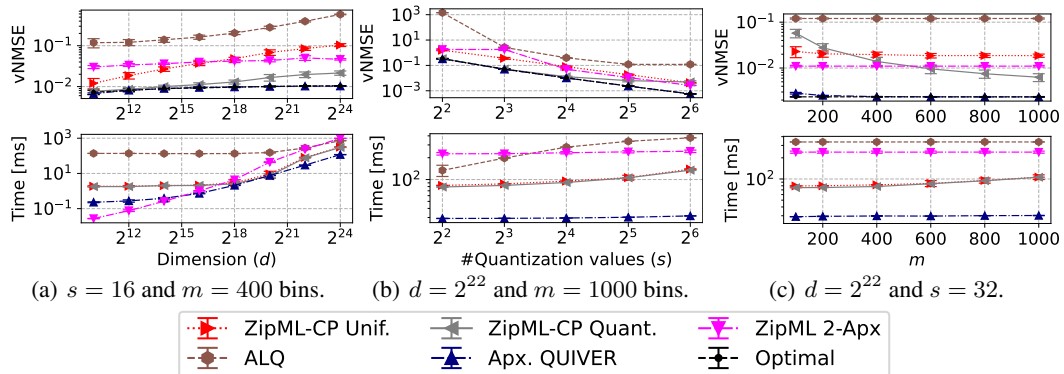

Figure 12: Comparing approximate solutions with Weibull$(1, 1)$ distributed input.

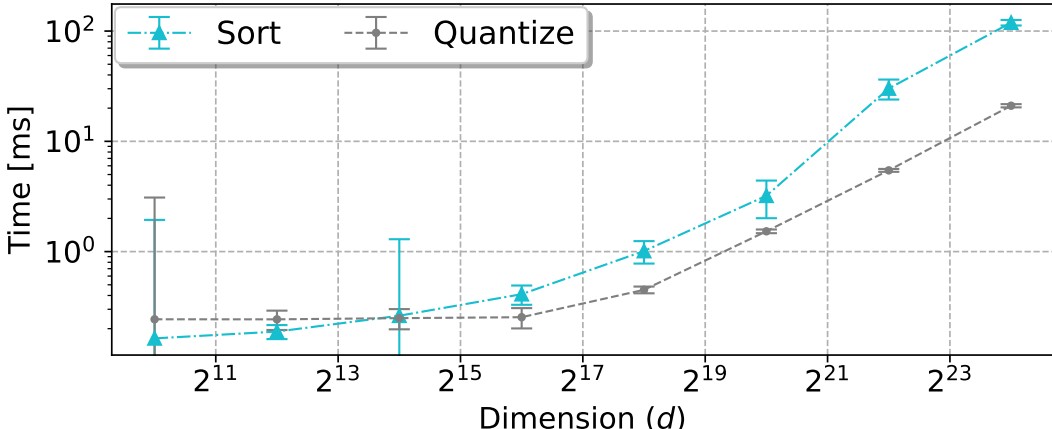

Figure 13: Sort and quantization times ($s = 16$) vs. $d$ on a T4 GPU.

