# OpenReview forum: "Optimal and Approximate Adaptive Stochastic Quantization"
_NeurIPS.cc/2024/Conference — NeurIPS 2024 poster_

### Official Review · Reviewer_SB91 · 2024-06-30

**Soundness:** 3
**Presentation:** 3
**Contribution:** 3
**Rating:** 7
**Confidence:** 2

**Summary:**

The paper is concerned with quantization,
that is encoding the components of a vector $X\in \mathbb{R}^{d}$ in a finite
alphabet $Q$ of given size $s$. The goal is unbiased,
stochastic quantization, where for a given component $x$ the encoding value $%
\hat{x}$ are chosen at random such that $E\left[ \hat{x}\right] =E\left[ x%
\right] $. Given $Q$ this fixes the encoding procedure, so that the
principal problem is the choice of the alphabet $Q$, which should minimize
the mean squared error (MSE) $E\left[ \left\Vert X-\hat{X}\right\Vert \right]
$, for fixed size $s$. It is important to realize that the optimal alphabet
is a subset of the set of components of $X$, so $s\leq d$. An optimal
solution has been found by Zhang et al [24] and the same paper already
offers an improved algorithm called ZipML of $O\left( sd^{2}\right) $
runtime and $O\left( d^{2}\right) $ memory requirement.

The paper at hand builds on the analysis of the latter paper, and gives an
improved algorithm  of $O\left( sd\right) $ both runtime and memory
complexity. Using a closed form solution for $s=3$ runtime and memory
requirements are halved in another improvement of the algorithm. Just as the
previous algorithm this version also returns the optimal solution.

To further accelerate the method the authors discretize the quantization
values on a grid of fixed size $m+1>s$ and seek a subset of this grid to
minimize the MSE for resulting set. The algorithm they present then has
space and time complexity of $O\left( d+ms\right) $. Because of this
improvement on can afford a larger cardinality of the set of quantization
values. The excess error of this approximate algorithm for $2s-2$
quantization values over the previous algorithm with $s$ quantization values
is bounded.

The paper then gives results of numerical experiments with various
distributions.

**Strengths:**

The paper considers a problem of obvious practical relevance.

The background and the various solutions are presented in a clear way,
easily understandable even to me, who has never very much considered the
quantization problem in this form.

I checked most of the analysis which appeared correct.

**Weaknesses:**

I cannot identify any major weaknesses. I must admit, however,
that, not being an expert in this field, I cannot judge if any other relevant
literature on unbiased, minimum-variance quantization beyond [24] is missing.

**Questions:**

I am a bit bewildered about calling $X$ a vector, since the ordering is appearantly irrelevant. Also multiple values do not change anything. "Set of real numbers" seems more appropriate to me.

**Limitations:**

In a subsection "limitations" the authors admit that their algorithm is not GPU-friendly. They also remind the reader that the algorithm requires initial sorting of the data.

---

> ### Author Rebuttal · Authors · 2024-08-05
>
> Thank you for the review.
>
> > I am a bit bewildered about calling $X$ a vector, since the ordering is appearantly irrelevant. Also multiple values do not change anything. "Set of real numbers" seems more appropriate to me.
>
> We agree that the input can be considered as a multiset (and not a set - a counter-example is given below) of entries we wish to quantize. Our solution, however, is based on a dynamic program that looks at prefixes of the vector one gets by looking at the sorted version of this multiset.
>
> Why $X$ cannot be modeled as a (simple) set:
>
> For example, if the entry `5’ appears twice, it carries double the weight when considering the sum of variances.
>
> More concretely, consider the input $X=(0, 3, 5, 5, 7)$ and $s=3$. The optimal solution in this case is $Q=\set{0,5,7}$. In contrast, for the input $X=(0, 3, 5, 7)$, the optimal solution is $Q=\set{0, 3, 7}$, giving a sum of variances of $4$, compared to a sum of variances of 6 if using $Q=\set{0, 5, 7}$ instead.
>
>
> > I cannot identify any major weaknesses. I must admit, however, that, not being an expert in this field, I cannot judge if any other relevant literature on unbiased, minimum-variance quantization beyond [24] is missing.
>
> Please note that we also compare our approach with ALQ [26]. We also cite other works (e.g., [25]) that look at special cases (a specific distribution) of the ASQ problem, but are not addressing the general formulation.

---

> > ### Comment · Reviewer_SB91 · 2024-08-12
> >
> > Thank you for the clarification provided. I will keep my score, modulo potential insights in the discussion phase with the other reviewers.

---

### Official Review · Reviewer_KyKu · 2024-07-10

**Soundness:** 3
**Presentation:** 3
**Contribution:** 3
**Rating:** 6
**Confidence:** 3

**Summary:**

The paper presents an algorithm to solve the Adaptive Stochastic Quantization (ASQ) problem that is claimed to be more computationally efficient than existing solutions. The paper also presents simulations showing the improved efficiency.

I have read the authors' responses to my comments and made changed the overall score.

**Strengths:**

The paper is well-written.
There is sufficient background material on previous works on the topic.
The simulation study is clear, appears comprehensive, and there is informative discussion.

**Weaknesses:**

The motivation for ASQ is unclear to me. In a discussion about quantization in the form of lossy compression in the context of NeurIPS, I’d expect to learn at minimum: (1) what is the source of redundancy that makes quantization possible without affecting too much the performance. (2) How the method exploits this redundancy. The statements in lines 67-71 appear to say that the authors do not care about these aspects.
Specifically to ASQ:
In the background, the authors nicely explain the benefit of adaptivity and unbiasedness, but do not explain the benefit of stochasticity. For example, what is the benefit of introducing randomness of the proposed type compared to the classical Lloyd algorithm (Lloyd 1982 https://ieeexplore.ieee.org/document/1056489)?

I understand that ASQ has been studied before, which might be interesting, and thus the contribution of the paper might be significant. However, in my opinion, it is not a good fit for NeurIPS since the contribution is only associated with computations/implementation of a known quantization method.

Another weakness is that the ASQ problem is formally stated only in lines 103-104. The problem appears simple enough to be presented in the first few lines of the paper.

Other comments:
Line 106: the inclusion of Q in X seems a typo.

**Questions:**

Please state the problem much earlier.

**Limitations:**

The paper addresses the limitations appropriately.

---

> ### Author Rebuttal · Authors · 2024-08-05
>
> Thank you for the review.
>
> > What is the source of redundancy that makes quantization possible without affecting too much the performance?
>
> The source of redundancy depends on the ASQ use case. Here are two specific examples, while other use cases may have a different motivation.
>
> *Example 1: Gradient compression (GC).* (Lines 81-87 in our submission)
>
> GC is a common building block for distributed and federated learning where multiple workers (clients) participate in the learning process. At each round, workers compute their gradients and then aggregate them in order to update the model. Since the process has inherent variance (each such stochastic gradient is computed with respect to a random subset of the data), one can leverage compression to alleviate communication bottlenecks without a significant impact on the accuracy of the aggregated (global) gradient if the variance of the compression is small compared with this inherent variance.
> Moreover, as recent works show, it is important for the compression to be unbiased. Intuitively, when the compression is unbiased (and independent among workers) some workers round up and some round down, allowing the error of the average to decrease proportionally to the number of workers (in expectation).
>
> *Example 2: Model compression (MC).* (Lines 88-92 in our submission)
> In MC, one quantizes the weights of a model in order to decrease its space requirements and improve the inference time. In this use case as well, it has been demonstrated that applying biased methods like Round-To-Nearest to compress parameters of large language models can lead to worse results compared to stochastic quantization. This is due to the reliance on the parameters of LLM layers for calculating inner products with their inputs. Ensuring these inner products remain unbiased results in less error in the outputs of the layers and less bias accumulation among layers, thereby enhancing overall accuracy.
>
> For both examples, we cite works that substantiate these claims.
>
> > How the method exploits this redundancy. The statements in lines 67-71 appear to say that the authors do not care about these aspects. Specifically to ASQ: In the background, the authors nicely explain the benefit of adaptivity and unbiasedness, but do not explain the benefit of stochasticity. For example, what is the benefit of introducing randomness of the proposed type compared to the classical Lloyd algorithm (Lloyd 1982 https://ieeexplore.ieee.org/document/1056489)?
>
> Lloyd’s algorithm is a great solution for **when the quantization is allowed to be biased**. In fact, *it is impossible to be unbiased without stochasticity* when quantizing since multiple inputs can be quantized to the same value.
>
> Moreover, while Lloyd’s algorithm may not yield the optimal solution, we compare with the **optimal** biased algorithm (for 1D, there is an algorithm that finds the optimal solution, see https://arxiv.org/abs/1701.07204).
>
> That is, the optimal biased algorithm shown in Figure 1 is at least as accurate as Lloyd’s. When estimating a single vector, it is indeed more accurate than unbiased methods, but as we explain, it is unsuitable for cases when unbiasedness is desired, e.g., when averaging the sum of multiple independently compressed vectors as in federated learning settings.
>
> > I understand that ASQ has been studied before, which might be interesting, and thus the contribution of the paper might be significant. However, in my opinion, it is not a good fit for NeurIPS since the contribution is only associated with computations/implementation of a known quantization method.
>
> We politely disagree. The known quantization method is considered impractical for many use cases (see lines 43-48), and this work enables the use of ASQ for multiple ML applications, as shown in these previous works.
>
> Moreover, the ASQ problem is well-established and of high interest to the ML community and NeurIPS in particular, as the line of related works we cite indicates (e.g., [24] and [26], which we compare with, were published in ICML and NeurIPS).
>
> > Another weakness is that the ASQ problem is formally stated only in lines 103-104. The problem appears simple enough to be presented in the first few lines of the paper.
>
> Thank you for the suggestion. We will expand in lines 19-21 to explain that the goal is to find such a set of quantization values $Q$.
>
> > Other comments: Line 106: the inclusion of $Q$ in $X$ seems a typo.
>
> This is not a typo. The meaning of $Q\subseteq X$ is that there exists an optimal solution in which all quantization values are a subset of the input entries. This is a known observation that was mentioned, e.g., in [24]  (as we mention in Line 106).
> We will explain this further in the text.
>
> > Please state the problem much earlier.
>
> Will do!

---

> ### Comment · Reviewer_KyKu · 2024-08-13
>
> Thank you for addressing my comments.
>
> I have changed my mind about the fit to NeurIPS.
>
> Concerning the discussion about the source of redundancy, perhaps there are similarities with the unfolding/unrolling optimization concept. For example, see:
> Monga, Vishal, Yuelong Li, and Yonina C. Eldar. "Algorithm unrolling: Interpretable, efficient deep learning for signal and image processing." IEEE Signal Processing Magazine 38.2 (2021): 18-44.

---

> > ### Author Response · Authors · 2024-08-13
> >
> > Thanks for the reference!
> > We will study it and discuss it in the paper.

---

### Official Review · Reviewer_QWUX · 2024-07-16

**Soundness:** 3
**Presentation:** 1
**Contribution:** 3
**Rating:** 4
**Confidence:** 2

**Summary:**

This paper studies the Adaptive Stochastic Quantization (ASQ) problem and presents a dynamic programming based algorithm that improves time and space complexities.

**Strengths:**

The algorithm is compared with peer dynamic programming based algorithm called ZIPML. Quiver Algorithm introduced in this paper is showing great  potential to be used for  compression of ML models. The algorithm has the advantage of controllable tradeoff between
accuracy and speed.

**Weaknesses:**

The presentation of the paper needs  further improvement. please proofread the paper for  grammatical errors and  typos, some of  which I point out  to here:

ln. 29, $X’$

on ln37,   $\hat{X}$ concatenation of  $\hat{x}$’s ?  define all  parameters before using them.

same for $\theta(s)$ on ln 107.

-ln. 106   is Q in X ?

- Many abbreviations are undefined: ALQ,QSGD,RTN .
- ln. 56-57, It is  ambiguous  to which algorithms you are comparing  the  time and space  complexity of your algorithm.
- what does  “on a commodity PC” mean? provide specifications  of your processor.
- ln. 80 "orders of magnitude lower error" , compared to which algorithm?
- The definition of MSE[i,j] in 109 is not clear. please write it mathematically.

- The main problem  of the paper which is defined in on ln.116 is not very clear. No description or references are provided. The authors are leaving many foundational definitions of dynamic programing and its application to quantization up to the reader. The paper better be inclusive of essential  concepts.

**Questions:**

you begin your paper by the assumption of  1-bit quantization  i.e.  $x \in \{a,b\} $(another typo in the paper). However the results are implemented for   more number of bits such as s=16. Is there any reason  you made 1-bit assumption on ln 94?

**Limitations:**

The proposed Algorithm is not implemented for ML model and  gradient compression.

---

> ### Author Rebuttal · Authors · 2024-08-05
>
> Thank you for the review.
>
> Thanks for pointing out the typos, we will fix these.
>
> > In 106 is Q in X ?
>
> We write that there exists an optimal solution for which $Q\subseteq X$ (and credit [24] for this observation). That is, there exists an optimal solution where all quantization values are entries in the input vector $X$.
>
> > Many abbreviations are undefined: ALQ,QSGD,RTN .
>
> ALQ and QSGD are the names of the algorithms. They were coined by the respective authors of [26] and [13]. The full name of the acronym RTN is given in line 78.
>
> > ln. 56-57, It is ambiguous to which algorithms you are comparing the time and space complexity of your algorithm.
>
> This is compared to the state of the art. As explained later (line 121), this is compared to ZipML [24]. We will make that clearer in the introduction.
>
> > what does “on a commodity PC” mean? provide specifications of your processor.
>
> Please see lines 231-233.
>
> > ln. 80 "orders of magnitude lower error" , compared to which algorithm?
>
> Compared to the non-adaptive algorithms, as we point out the benefits of adaptivity.
>
> > The definition of MSE[i,j] in 109 is not clear. please write it mathematically.
>
> Recall that $a_x = \max \set{q\in Q\mid q\le x}$, $b_x = \min \set{q\in Q\mid q\ge x}$, and that $X_j$ is the vector with the first $j$ entries of $X$.
>
> The mathematical formulation is:
> $$
> MSE[i,j] = \min_{Q: |Q|\le j, x_j\in Q} \sum_{x\in X_j} (b_x-x)(x-a_x).
> $$
>
> We will add this to the paper.
>
> > The main problem of the paper which is defined in on ln.116 is not very clear. No description or references are provided. The authors are leaving many foundational definitions of dynamic programing and its application to quantization up to the reader. The paper better be inclusive of essential concepts.
>
> We kindly disagree. Dynamic programming is a basic tool in computer science and the specific problem is explained (lines 117-120 give the semantic meaning of the parameter $k$ and MSE[i,j] was defined above).
>
>
> > **Q1:** you begin your paper by the assumption of 1-bit quantization i.e. $x\in a,b$
> (another typo in the paper). However the results are implemented for more number of bits such as s=16. Is there any reason you made 1-bit assumption on ln 94?
>
> We do not assume that $x\in\{a,b\}$.
>
> In stochastic quantization, for some $a,b\in\mathbb R$, the input value $x\in[a,b]$ is quantized to $\widehat x\in \{a,b\}$. That is, the input is a real value between $a$ and $b$, and the quantized value is one of $a,b$.
> When quantizing a vector $X$ using $s>2$ quantization levels, each entry $x\in X$ is stochastically quantized between the two encompassing values $a_x,b_x$ as explained in lines 101-102.
>
> > **L1:** The proposed Algorithm is not implemented for ML model and gradient compression.
>
> Since QUIVER solves the same problem as previous solutions such as ZipML, there is no gain in recreating experiments that would yield the exact same results. As you can observe from vNMSE subfigures of Figure 2, both QUIVER and ZipML have identical error, as they provide the optimal solution for the ASQ problem. The benefit of QUIVER is in its faster runtime and lower space requirements, as we evaluate in the remaining figures.
>
> Moreover, our paper's focus is not on model or gradient compression but on improving the state of the art for the well-established ASQ problem, which is of high interest to the ML community and NeurIPS in particular, as the line of related works we cite indicates (e.g., [24] and [26], that we compare with, were published in ICML and NeurIPS).

---

> > ### Comment · Reviewer_QWUX · 2024-08-12
> >
> > Thank you for clarification of some of my points. Some are unaddressed in your rebuttal.  Please add clear definition of  MSE[i,j] to the paper. The one I see in your rebuttal comments does not show  dependency on i. You can add a notation definition subsection to avoid confusions, for instance \hat{x} \in \{a_x,b_x\}  is your notation for ASQ, please don't assume the readers know it already. The ASQ is the main definition in your paper, yet you defined it only inline on ln. 94-97. It better be a numbered equation to  be clear.  I see other reviewers commenting on some notations as well. I still think the presentation of the paper needs much further improvement, therefore I will keep my score.

---

> > > ### Author Response · Authors · 2024-08-13
> > >
> > > Please read the submission text (lines 109-110), which is comprehensive and correct:
> > >
> > > ``we denote by $MSE[i,j]$ the optimal MSE of quantizing the prefix vector $X_j=\langle{x_1,\ldots,x_j}\rangle$ **using $i$ quantization values** *that include* $x_j$''.
> > >
> > > That is, the corresponding mathematical formulation is:
> > > $$
> > > MSE[i,j] = \min_{Q: |Q|\le {i}, x_j\in Q} \sum_{x\in X_j} (b_x-x)(x-a_x).
> > > $$
> > >
> > > We are puzzled by your low score, which is due to minor typos and presentation issues, without any evaluation of the contribution of the proposed methods.

---

> > > > ### Comment · Reviewer_QWUX · 2024-08-13
> > > >
> > > > Thank you for correcting the terms. Although the presentation of the paper should be clear enough to not raise these concerns, I will defer to the other reviewers' comments and area chair.

---

### Official Review · Reviewer_HN3u · 2024-07-16

**Soundness:** 3
**Presentation:** 3
**Contribution:** 3
**Rating:** 6
**Confidence:** 3

**Summary:**

The paper proposes the QUIVER algorithm, Accelerate QUIVER (an accelerated variant when s = 3), and Apx QUIVER (a variant that utilizes approximations for better speed) to solve the Adaptive Stochastic Quantization (ASQ) problem. The paper also provides theoretical guarantees to their algorithm, improving the current SOTA time complexity from $O(s d^2)$ to $O(s d)$ and space complexity from $O(d^2)$ to $O(s d)$. Practical experiments are also run to show the improvement of the current SOTA.

**Strengths:**

1. Technically solid: The technical aspects of the paper such as formulations and proofs are clear and correct.
2. Clear motivations: The benefits of adaptivity and unbiasedness are explained clearly with practical illustrations of the MSE reduction of different methods.
3. Theoretically and practically outperform SOTA: The paper makes it clear in the contributions and experiments sections how the proposed algorithms outperform past methods practically and with better theoretical complexities.
4. Weaknesses acknowledgements: The authors clearly state and discuss the drawbacks of their algorithms.

**Weaknesses:**

1. SMAWK algorithm seems to be an important subroutine in QUIVER, I think it should be elaborated a bit more for those who are not aware of SMAWK and how the process takes O(d) time and space (calls of SMAWK are referred plentifully in the paper).
2. Following up on that, I also wonder if the authors can elaborate for me on the key novelty of QUIVER since the key update step in this algorithm comes from SMAWK (is it the preprocess?). I do see more clearly the novelty of Accelerated QUIVER with the important observation when s = 3 (also related to question number 1 below).
3. While Accelerated QUIVER offers some interesting ideas, it does not work for s > 3. I believe the section can be stronger if the authors can maybe elaborate some (common) situations or (useful) applications for s = 3 to contextualize the “practicality” of this algorithm.

**Questions:**

1. Are there alternatives to SMAWK that can be implemented (with some small modifications) that still allow QUIVER to work?
2. The author states that Apx. QUIVER does better in practice than the bound in Lemma 6.1, just out of curiosity, what do you think is the reason?
3. Are there any other results in the literature that you can compare with Apx. QUIVER results (such as line 223 - 224)?
4.  The asymptotic difference among the approximate algorithms are not clear to me in Figure 3, can the author elaborate further?

**Limitations:**

Yes

---

> ### Author Rebuttal · Authors · 2024-08-05
>
> Thank you for the detailed review. We first address the questions in detail and *relate to the weaknesses in the comment that follows*.
>
>
>
> **Q1** While SMAWK has the optimal $O(d)$ time complexity for the problem, there are simpler approaches. For example, by leveraging the fact that if MSE[i,j] is minimized at k, then for all j’>j, MSE[i,j] must be maximized for some k’>k. This allows a `binary search approach’ that solves the problem with a slower $O(d \log d)$ time complexity. (This approach is standard and well-known.)
> We implemented this approach and empirically verified that the resulting solution is slower than using SMAWK.
>
> **Q2** The analysis is (asymptotically) tight only for the worst-case but is far from tight for non-adversarial inputs. Namely, we prove the theoretical claim by constructing a specific solution with $2s-2$ quantization values that is derived from the optimal solution with $s$ quantization values. This solution is generally far from the optimal solution when using $2s-2$ quantization values, even when restricted to grid entries.
>
> The intuition is as follows: we placed ($2s-4$) of the quantization values in consecutive pairs on the grid (e.g., $\ell=17$ and $\ell=18$). In practice, Approx. QUIVER finds the optimal solution with quantization values that are on the grid, and such a solution is unlikely to contain many pairs of consecutive quantization values (for realistic input vectors). A secondary reason is that the worst-case analysis considers that all coordinates might be in the middle of two grid points, which is unlikely in non-adversarial inputs.
>
> **Q3** Are you referring to [16]? As we discuss in lines 26-29, this is an example of a line of work that optimizes the quantization error for the worst-case inputs by applying a transformation on the input vector. Also, as we explain in lines 67-71, such works are orthogonal to ASQ since one can apply it after the transformation, whereas [1, 3, 4, 15, 16, 17] use pre-computed quantization values instead of the ones that are optimal for the transformed vector. That is, by applying adaptive quantization to select the quantization values after the transformation, we can get a solution with improved accuracy at the cost of additional computation.
>
> We further note that comparing worst-case solutions to adaptive ones (without applying the transformations) is not an apples-to-apples comparison. Namely, if the input has structure, ASQ can provide a significantly more accurate solution. On the other hand, if the input is distributed adversarially, no benefit can be attained by adaptive algorithms.
>
> To see the benefit of adaptive solutions over worst-case solutions, consider a trivial example where the input vector has $d/2$ (+1) entries and $d/2$ (-1) entries. ASQ trivially solves this with zero error even for $s=2$. In contrast, all the above algorithms will apply a transformation after which the structure is gone, and any quantization will have a significant error.
>
> **Q4** Thanks for pointing out that this is not explained. We will clarify this in the paper.
> Namely, we compare with several approximate algorithms:
>
> *ZipML-CP* [24] is an algorithm that runs the exact ZipML algorithm on a subset of the points called `Candidate Points’. Since ZipML runs in $O(d^2 s)$ time, here we use M candidate points to get  $O(d + M^2 s)$ time.
>
> *ZipML 2-Apx* [24] is an algorithm that computes an approximate solution in $O(d \log d + s^3)$ time.
>
> *ALQ* [26] is an algorithm that finds good quantization values for a truncated normal distribution. To use it, it samples several gradients (by computing the gradient of several random batches) to fit the truncated normal parameters. To be fair to ALQ, since we evaluate a single-shot quantization scenario, we calculate the input vector's exact mean, variance, and support parameters. Yet, its algorithm for finding the quantization values for the distribution, which uses an alternating coordinate descent method, underperforms for two main reasons: (1) the distribution, generally, is not truncated-normal. (2) In each iteration of the coordinate, they optimize the location of each quantization value i by fixing the locations of the (i-1)’th and (i+1)’th values and calculating the optimal location by integrating over the distribution truncated to this range.
>
> This then runs for several (we used 10, as in their released code) iterations, so in total, they compute $\approx 10s$ integrals. While theoretically requiring $O(d)$ time, in a model where such integral calculation takes constant time, this is markedly slower than other approaches. We note that it is possible that with low-precision integral calculations, one may improve the runtime, but the error (which is already not competitive) will degrade further.
>
> **We will relate to the weaknesses in the following comment.**

---

> ### Author Response · Authors · 2024-08-05
> **Relating to the Weaknesses**
>
> **W1** Thank you for the suggestion. We agree that explaining the SMAWK algorithm in detail and providing more intuition about it will improve the paper, and we will add these for the camera-ready version.
>
> We provide the high-level details here:
>
> *SMAWK Algorithm Steps*
>
> * Pruning Phase:
> > Remove columns that cannot possibly contain a row maximum. This is done by comparing each column with its neighbors and discarding those that cannot be maxima based on the totally monotone property. At the end of this phase, the number of columns can be no larger than the number of rows.
>
> * Recursive Reduction:
> > The algorithm reduces the problem size by considering a subset of the rows and columns. It selects every other row and recursively solves the reduced problem.
>
> * Candidate Set:
> > After solving the smaller problem, the solution provides candidate columns for the original problem. The algorithm only needs to consider these columns to find the maxima for the skipped rows.
>
> * Merge Phase:
> > Combine the results from the reduced problem with the candidate set to find the maximum for each original row.
>
> ---
>
> **Efficiency:**
> The SMAWK algorithm achieves a time complexity of $O(d)$ for a $d×d$ matrix. This efficiency is due to the recursive reduction of the problem size and the properties of totally monotone matrices that limit the number of comparisons needed. Namely, the pruning step takes $O($#$cols)$, where #$cols$ is the number of columns still being considered. The crux is that the recursive step happens after the pruning, which means that the recursive invocation happens with a number of columns that is, at most, double the number of rows (as the number of rows is halved). This means that the overall complexity of each recursive step is proportional to the number of rows, yielding the recursion:
> $T(n) = T(n/2) + O(n) = O(n)$.
> A simple example Python (by David Eppstein) implementation appears here: https://github.com/pombredanne/code-5/blob/master/recipes/Python/117244_SMAWK_totally_monotone_matrix_searching/recipe-117244.py.
> Our implementation is in optimized C++ and we will release it as open source with the publication of the paper.
>
> **W2**  The novelty of QUIVER is the identification that producing MSE[i,·] from MSE[i-1,·] can be expressed as the problem of finding the maximas in an *implicitly defined* matrix. This requires several steps, including our pre-processing (to enable matrix queries in constant time) and proving the quadrangle inequality. Further, finding MSE[i,·] from MSE[i-1,·] is only a subroutine in the algorithm which invokes it several times and also reconstructs the resulting solution.
>
> Further novelty is in our Accelerated QUIVER algorithm which provides a faster solution to the same problem, and the Approx. QUIVER that provides further speedup at the cost of a small error.
>
> We will further clarify our contribution in the paper.
>
> **W3** Accelerated QUIVER works and is faster than QUIVER for all $s$. When $s>3$, it also requires invoking the SMAWK algorithm as a subroutine, but the number of invocations reduces significantly.  We will further clarify this in the text.
>
> Namely, we provide a closed-form, computable in constant time, solution for the $s=3$ problem, and then reformulate the dynamic program (see lines 171-182) to allow fewer recursive steps, thereby improving both the speed and space requirements both theoretically and in practice.
>
> If our understanding of this weakness is incorrect, we would appreciate if the reviewer could clarify.

---

> > ### Comment · Reviewer_HN3u · 2024-08-08
> > **Reviewer Response**
> >
> > I really appreciate the detailed answers and the responses the author made regarding my reviews. Most of my points are addressed properly, and I believe the revised version with the additional details will be quite solid. While I will have to see further discussion with AC and other reviewers to improve the score I have given, I am certainly in support of this paper acceptance.

---

### Decision · Program_Chairs · 2024-09-25

**Decision:**

Accept (poster)

**Comment:**

The authors have studied quantization as a map from a multi-dimensional real space to a finite alphabet of a given size. The authors focus on unbiased quantization and the mean squared error loss and improved the solution by Zhang et al [24] in terms of runtime and memory complexity.

The paper was thoroughly discussed and the reviews are mixed. The reviewers are positive about this work in general, and they have found concerns such as a thorough literature review beyond [24]  and the key novelty of QUIVER considering SMAWK. I think the presentation of the paper should be improved significantly, but these can be improved for the revised manuscript. I would recommend that for the revised version, the authors take into account the comments raised by reviewers and improve the paper as they have promised in the rebuttals.